Exosomes in cancer: small vesicular transporters for cancer progression and metastasis, biomarkers in cancer therapeutics

Abak Atefe atefeh.abak@gmail.com abaka@tbzmed.ac.ir 1
Abhari Alireza 2
Rahimzadeh Sevda 2
1 Department of Medical Genetics, Faculty of Medicine, Tabriz University of Medical Sciences , Tabriz , Iran
2 Department of Biochemistry and Clinical Laboratory, Faculty of Medicine, Tabriz University of Medical Sciences , Tabriz , Iran
Sanderson J. Thomas
Electronic publication date: 2018 May 29
Publication date: 2018
Volume: 6
Electronic Location ID: e4763
Received 2018 Feb 22; Accepted 2018 Apr 23
Copyright: ©2018 Abak et al.
Copyright year: 2018
Copyright holder: Abak et al.
License: This is an open access article distributed under the terms of the Creative Commons Attribution License, which permits unrestricted use, distribution, reproduction and adaptation in any medium and for any purpose provided that it is properly attributed. For attribution, the original author(s), title, publication source (PeerJ) and either DOI or URL of the article must be cited.
License URL: https://creativecommons.org/licenses/by/4.0/

Keywords: Cancer, Exosome, Angiogenesis, Metastasis, Therapy

Funding: Tabriz University of Medical Sciences This work was supported by Tabriz University of Medical Sciences. The funders had no role in study design, data collection and analysis, decision to publish, or preparation of the manuscript.

==============================
Cancer progression is a polygenic procedure in which the exosomes can function as substantial roles. Exosomes are tiny, phospholipid bilayer membrane nanovesicles of endocytic derivation with a diameter of 40–100 nm. These nanovesicles can transport bioactive molecules containing mRNAs, proteins, DNA fragments, and non-coding RNAs from a donor cell to recipient cells, and cause the alteration in genetic and epigenetic factors and reprogramming of the target cells. Many diverse cell types such as mesenchymal cells, immune cells, and cancer cells can induce the release of exosomes. Increasing evidence illustrated that the exosomes derived from tumor cells might trigger the tumor initiation, tumor cell growth and progression, metastasis, and drug resistance. The secreted nanovesicles of exosomes can play significant roles in cells communicate via shuttling the nucleic acid molecules and proteins to target cells and tissues. In this review, we discussed multiple mechanisms related to biogenesis, load, and shuttle of the exosomes. Also, we illustrated the diverse roles of exosomes in several types of human cancer development, tumor immunology, angiogenesis, and metastasis. The exosomes may act as the promising biomarkers for the prognosis of various types of cancers which suggested a new pathway for anti-tumor therapeutic of these nanovesicles and promoted exosome-based cancer for clinical diagnostic and remedial procedures.

Introduction

The solid tumors are complicated structures that including the surrounding tumor stroma and cancer cells, which composed of endothelial cells, fibroblasts, and immune cells (Sund & Kalluri, 2009). The surrounding cells are permanently extracting factors that alter the tumor microenvironment (TME) directly or indirectly (Dvorak et al., 2011). A persistent cross-talk among tumor cells and the distant tumor microenvironment have applied as the pivotal tumor growth, and significant targets in antitumoral intervention, and systemic diffusion (Kalluri & Zeisberg, 2006; Swartz et al., 2012). Extracellular vesicles (EVs) have appeared as long-distance communicators; their outcomes in primary tumors can display as systemic effects and contribute to procedures within the circulation by many various kinds of cells. The exosomes pretense a special class of EVs, which released via various kinds of cells (Desrochers, Antonyak & Cerione, 2016; Kowal, Tkach & Théry, 2014; Mathivanan, Ji & Simpson, 2010). Newly evidence represents that the release of exosomes has been detected to play a considerable role between human tumor cells and systemic cell-to-cell relevance in cancers. The exosomes, initially defined through several common traits in reticulocytes three decades ago, containing morphology (a classic “dish” or “cup” formed in transfer electron microscopy (SEM)), density (1.13–1.19 g/ml), size (30–100 nm in diameter), and determined increased protein markers (TSG101, HSP70, and tetraspanins) (Harding, Heuser & Stahl, 2013). Recent publications illustrated that exosomes are small membrane nanovesicles shaped in multivesicular bodies of endocytic derivation with a diameter of 40–100 nm. Exosomes were primarily considered as the trash bags for elimination of abandoned membrane segments and unwanted molecular fragments from cells, besides the critical task of exosomes in stimulation of immune response is identified as their effect on antigen presentation in the mid-1990s (Raposo et al., 1996). Interestingly, the scholars detected that noncoding RNAs (miRNAs), messenger RNAs (mRNAs), proteins and DNA fragments could be burdened as “goods” in extracellular vesicles (EVs) (Balaj et al., 2011). Likewise the exosomes as a nanovesicles were detected to function as “communication shuttles” from a donator cells to recipient cells, that could able to re-encode genes of receiver cells, reprogramming of the tumor microenvironment and recruitment to shape a pro-tumorigenic soil, and play a considerable act in the progression, invasion, metastasis, and become insensitive to a drug of cancer (Azmi, Bao & Sarkar, 2013; Balaj et al., 2011; Valadi et al., 2007).

Here we reviewed a new science concerning the function of exosomes as a shuttle in tumorigenesis, emphasizing their biogenesis, component, significant affection, and then considering the potential of exosomes as a novel biomarkers for clinical remedial target diagnosis and prognosis.

Survey Methodology

PubMed was mostly utilized to search for relevant articles published utilizing the keyword “exosome”, “cancer” and “therapy.” Afterward, screened articles were utilized as references for this review. Additional keywords, such as “microenvironment,” “nanovesicles” and “tumor,” were also utilized.

Exosomes Biogenesis, Release, and Uptake

Contrary to the larger microvesicles (MVs), that straightly shed from the cell membrane, the exosomes forming is a specific process that contains four steps: beginning, endocytosis, multivesicular bodies (MVBs) creation, and finally the exosome secretion (Théry, Zitvogel & Amigorena, 2002). Exosomes primarily can shape through the ceramide-induced procedures of inside budding from the late endosome restricted membranes (Trajkovic et al., 2008). The encapsulation of RNA molecules and functional proteins occur through this process. The Multivesicular bodies (MVBs) within the endocytic systems shaped via the budding of an endosome limited membrane into the extracellular milieu of the section by the junction and merge of the MVBs with the cell membrane. The MVBs are either classified as the destroying of cargo in the lysosome or leading to secretion within the extracellular space as exosomes after vesicular cumulation (Février & Raposo, 2004; Trams et al., 1981). The procedures based on the classified of exosomal cargo within the intraluminal vesicles (ILVs) are still not completely understood. Although it has been offered to characterize the exosomes formation and releasing by both endosomal tethering complexes necessitated for transport (ESCRT)-dependent and independent symptoms, however, alternative ways may also exist (Trajkovic et al., 2008). The ESCRT pathway discerns ubiquitination of membrane proteins and promotes their internalization within the multivesicular endosome (Wollert & Hurley, 2010). The mechanism for the microvesicles formation has been illustrated to regulate through the Syndecan heparan sulfate proteoglycans and their cytoplasmic adaptor syntenin (Baietti et al., 2012). The MVB trafficking and the secretion procedure of exosomes may be performed through the outside of exosomes and the microvesicles budding procedure or through multiple compositions of the endocytic machinery, containing the members of the Rab guanosine triphosphatase (GTPase) family (Rab11, Rab 27a, Rab 27b, Rab 35), elevated expression of heparanase, SNARES (soluble NSF attachment receptor), and cytoskeleton regulatory proteins (Azmi, Bao & Sarkar, 2013; Beach et al., 2014; Ostrowski et al., 2010; Pant, Hilton & Burczynski, 2012). A promoted dissemination of exosomes is critically was detected to be triggered via multiple kinds of stress, including alters in PH membrane, oxidative stress, shear stress, hypoxia, thermal alters, and radiation, besides through formation of ceramide, stimulation of sphingomyelinase and following the p53- adjusted protein tumor-suppressor-activated pathway 6 (TSAP6) (Andaloussi et al., 2013; Azmi, Bao & Sarkar, 2013; Hannafon & Ding, 2013; Joanne et al., 2005; Kucharzewska & Belting, 2013; Lespagnol et al., 2008; Parolini et al., 2009; Yu, Harris & Levine, 2006). Exosomes shuttle information to the recipient cells via three major pathways: (1) interaction between receptor-ligand; (2) straight merge with cell membrane; (3) endocytosis through phagocytosis (Fig. 1). Also, there are multiple proteins that can function as specific receptors to activate the uptake of the exosome, containing ICAM-1 for APCs, and Tim 1/4 for B-cells (Miyanishi et al., 2007; Segura et al., 2005).

Figure 1 Schematic of exosomes derived cancer cell biogenesis and secretion.

Exosomes can secrete through cells while intracellular organs called multivesicular bodies (MVBs) fuse with the plasma membrane. The MVBs formation occurs through invaginations of late endosomes, which increased molecules from the Golgi apparatus (e.g., MHC class II molecules) or the cell surface (e.g., growth factor receptors). Subsequently, exosomes could be enriched in several materials including sphingomyelin, intracellular protein, ceramide, cholesterol, transmembrane receptors, mRNA, and miRNA. The exosomes secreted from human tumor cells can affect the local tumor microenvironment, alter the extracellular matrix, and enhance the angiogenesis, thrombosis and cancer cell proliferation.

Exosomes Structure and Composition

Exosomes are commonly cup-shaped extracellular small nanovesicles ranging in size from 30 to 100 nm diameter, consist of a phospholipid bilayer comprising membrane proteins that encircles a lumen containing an extensive range of biomolecules including carbohydrates, lipids, small fragments of DNA, mRNAs, proteins, and miRNAs inward to keep them from destruction (De Veirman et al., 2016; Hwang, 2013; Raimondo et al., 2011; Vlassov et al., 2012; Wang et al., 2016).

Protein composition of exosomes

Exosomes from several kinds of cells include a core set of similar proteins upon 4,600 various proteins have been related to these microvesicles, containing proteins from the phospholipid bilayer, endoplasmic reticulum, cytosol, and Golgi apparatus such as the heat shock proteins (HSP60, HSP70, and HSP90), the tetraspanin family (CD63, CD81, CD9, and CD82), cytoskeletal proteins (tubulin, moesin, actin, and syntenin), proteins involved in ESCRT complex (Alix, TSG101), phospholipases and lipid-related proteins (Bang & Thum, 2012; Hannafon & Ding, 2013; Harshman et al., 2013; Mathivanan, Ji & Simpson, 2010; Taylor & Gercel-Taylor, 2011; Vlassov et al., 2012).

Lipid composition of exosomes

Exosomes are containing a lipid bilayer including polyglycerophospholipids, sphingolipids (ceramide, sphingomyelin), raft-associated lipids (cholesterol), glycerophospholipids (phosphatidylethanolamine, phosphatidylserine, phosphatidylinositol, phosphatidylcholine), and phospholipids. Also, the lipid composition in exosomes varies significantly from that of the original cells (Record et al., 2014).

Nucleic acid composition of exosomes

The double-stranded deoxyribonucleic acids exist in these microvesicles derived from tumoral cells and cause the aberrant regulation of the derived cells (Silva et al., 2012). The other nucleic acids that carry with exosomes contain of mitochondrial DNA (mtDNA), messenger RNAs (mRNAs), microRNAs (miRNAs), long noncoding RNAs (lnc RNA), small-nuclear RNAs (snRNAs), small nucleolar RNAs (snoRNAs), piwi-RNAs (pi-RNAs), transfer RNAs (tRNAs), and ribosomal RNAs (rRNAs). Exosomal-containing RNA can transfer among a variety of cells and therefore is termed “exosomal shuttle RNA” (esRNA). Recently 764 miRNAs and 1639 mRNAs have been recognized in these nanovesicles from tissues of various species via a broad range of researches (Gajos-Michniewicz, Duechler & Czyz, 2014).

The composition of exosomes differs between various pathological and physiological status and originated cells. Also, the contents of these nanovesicles can discern from the derived cells because of the optional categorized of the cargo within exosomes.

Exosome Function

As a communicator, exosomes can directly shuttle the bioactive molecules among multiple kinds of recipient cells, with results in targeted cellular phenotyping, contained messenger-RNA (mRNA) and microRNA (miRNA) dependent on the shuttle of genetics, and also epigenetic information and lipid trafficking among cells (Azmi, Bao & Sarkar, 2013; Xiang et al., 2009). The existence of exosomes in the circulating body fluids reveals their role in various pathological situations, the instance of infection disease, cardiovascular disease, and progression of neurodegenerative disease (Bang & Thum, 2012; Fleming et al., 2014). The more substantial role of exosomes has been figured out in cancer which leads to tumor growth, angiogenesis, escaping from the immune response, causing tumor cell migration, stimulating normal cells to an invasion, and leading to metastatic colonization into distal tissues (Azmi, Bao & Sarkar, 2013).

Methods for the Isolation and Analysis of Biomarkers from Exosomes in Cancer Cells and Body Fluids

Recently, different methods are available for the isolation and discern of exosomes from the distinguished cells under normal and stressed situations, containing nucleic acid (DNA) sequencing, qRT-PCR analysis, western blotting assay, or Enzyme-linked immunosorbent assay (ELISA), which can identify RNA and protein of exosomes, also ultracentrifugation, source gradient ultracentrifugation combined with ultrafiltration centrifugation (SGUUC), commercial kits, magnetic activated cell sorting (MACS) can utilize as another method. To date, the international society for extracellular vesicles (ISEV) can apply for the detection of extracellular vesicles and their functions. Also, western blot and flow cytometry (FCM) are commonly utilized for recognizing of exosomes through discovering particular tetraspanins (for example CD9, CD63, and HSP70). Further, the transmission electron microscopy (TEM) can use for size and shape analysis (Lötvall et al., 2014). The common technique for the isolation of exosomes contains ultracentrifugation, which is often in combination with sucrose density gradients or sucrose cushions to float the relatively low-density exosomes. The ultracentrifugation procedure has multiple disadvantages: The manner is extremely labor-intensive and time-consuming; due to the restrictions of the design of ultracentrifuge rotors one cannot evaluate more than six specimens at a time; the procedure needs a major amount of raw materials; exosome productions are usually low; and vast training of staff is required (Théry et al., 2006; Zeringer et al., 2015). Isolation of exosomes based on size, by the prosperous isolation of exosomes through applying the ultrafiltration methods which are less time-consuming than ultracentrifugation and do not need the usage of the particular tool (Cheruvanky et al., 2007). HPLC (high-performance liquid chromatography)-based protocols could effectively permit the preparation of extremely pure exosomes. However, these methods need appropriative material and are not negligible to scale-up (Lai et al., 2010). Besides, the intricacy is that both body fluids and cell-culture media include an extensive amount of nanoparticles in the identical size range as exosomes. For instance, many miRNAs are included within extracellular protein complexes rather than exosomes (Wang et al., 2010). In addition, volume-excluding polymers such as polyethylene glycols (PEGs) could be used to precipitate exosomes from empirical specimens. The precipitate can be separated applying either low-speed centrifugation or filtration. System Biosciences presents an appropriative reagent called ExoQuick, which can be added to conditioned cell media, urine or serum, which precipitates these nanovesicles (Adams, 1973; Lewis & Metcalf, 1988; Yamamoto et al., 1970). In principle, a preferable resource for special purification of exosomes should be affinity isolation with antibodies to Alix, annexin, CD63, CD81, CD82, CD9, EpCAM, and Rab5. These antibodies could be collected on multiple media, containing microfluidic devices, plates, magnetic beads and chromatography matrices (Chen et al., 2010; Théry et al., 2006).

Exosomes and Cancer

Role of exosomes in cancerogenesis

As mentioned earlier, the exosomes revealed important roles in cancer progression. The exosomes released by human cancer cells are known as tumor-derived (TD) exosomes. The TD exosomes through autocrine signals can modulate the local growth progression of human cancer cells. The exosomal autocrine signaling pathway is related to kinds of cells and cellular traits, for instance, exosomes separated from gastric cancer cells with high CD97 (epidermal growth factor seven-transmembrane subfamily) expression enhanced cancer cell proliferation and invasion via exosome-mediated MAPK signaling pathway, and exosomal miRNAs may be contributing to induction of the CD97-associated pathway (Li et al., 2015). A mutant epidermal growth factor receptor (EGFRVIII), exists on the membrane of these nanovesicles originated from glioblastoma cells, can trigger cells loss of this mutant form. The integration of EGFRVIII within these cells caused by promotion of anti-apoptotic procedures and an augment in capacity for anchorage-independent growth (Al-Nedawi et al., 2008). On the other hand, the exosomes originated from pancreatic cancer cells enhance Bax expression, however, reduce Bcl-2 expression, cause the leading to cancer cells of the mitochondrial apoptotic pathway (Ristorcelli et al., 2008). This process illustrated that TD exosomes might act pivotal anti-cancer role through triggering apoptosis in several tumors. Accordingly, the determined or beneficial TD exosomes in vivo to their own survival relies on the cellular traits and kinds of the cells, which more research needs to be clarified. Moreover, the bone marrow mesenchymal stromal cells (BM-MSCs)-derived exosomes can support the multiple tumor cell expansion and development in various human cancer cells (Fig. 2).

Figure 2 Exosome recruitment of bone marrow-derived cells.

Exosomes transform the tumor microenvironment (TME) and dispose of distant tissue sites for metastasis. The efficacies of exosomes at distant tumor sites necessitate that exosomes migrate through the blood or lymph. They dispose tissue sites for metastasis or transform the bone marrow (BM) environment, and making a pre-metastatic niche to enhance tumor invasion and development. Thus tumor-derived exosomes can cause recruiting bone marrow-derived cells to the tumor and pre-tumor tissue where they function as cancer development and support the multiple tumor cell expansion and development in various human cancer cells.

Role of exosomes in tumor angiogenesis

The angiogenic procedures induced cancer cell progression can be activated through nutrient reduction, hypoxic, and in addition, inflammatory responses, generally detected in epithelial cell carcinomas. The neovascularization process from preexisting blood vessels associated with promoted endothelial cell proliferation, migration, and budding (Dvorak, 1986; Nazarenko et al., 2010). Vascular endothelial growth factors (VEGF), IL-8, transforming growth factor B (TGF-β), and fibroblast growth factor (FGF) are some of the angiogenic factors that function as endothelial cell proliferation and migration, can be necessary for the induction of tumor angiogenesis. Also, the exosomal miR-92a derived from leukemic cells can regulate integrin α5 to promote migration regulations and proliferation of endothelial cells and tube formation (Umezu et al., 2013). By other research, exosomes originated from melanoma cells including miR-9 were internalized through endothelial cells enhancing angiogenesis and metastasis via activation of the JAK-STAT pathway (Gajos-Michniewicz, Duechler & Czyz, 2014). Another report illustrated that CD-105-positive exosomes act an important role in establishing a niche in the lung microenvironment of SCID mice through the elevate expression of MMP2, MMP9, and VEGFR1 (Grange et al., 2011). In addition, the exosomes originated from hypoxic brain tumor glioblastoma multiform cells were increased with IL-8 and PDGF as angiogenic stimulatory molecules (Kucharzewska et al., 2013).

Role of exosomes in tumor metastasis

A major pathway in the metastatic cascade are tumor cell invasion and migration, missing the epithelial traits towards a more mesenchymal phenotype and the ability of the cell to attain a motile phenotype via changes in the cell to matrix interaction, disseminating tumor cells extravasate into remote sites and finally colonize secondary tissues and organs. There is an emerging report that shows tumor-derived exosomes are accomplished by tumor invasion and metastasis through regulating stromal cells, creating a pre-metastatic niche (Fig. 3), remodeling the extracellular matrix (ECM) and inducing angiogenesis (Alderton, 2012; Jung et al., 2009). Metastatic tumor cells dissemination enhanced level of miRNA by tumor-suppressor mechanism, that can indicate another procedure for the function of these nanovesicles in metastasis (Ostenfeld et al., 2014). The recent study illustrated that the exosomal proteins originated from tumor hypoxia of prostate cancer cells are associated with the process of adherens junctions in epithelial cells and cytoskeleton remodeling, including the enhanced metastasis and invasiveness in prostate cancer cells, is modulated through exosomes (Ramteke et al., 2015). Also, by recent investigate gastrointestinal stromal tumor cells (GISTs) secrete exosomes including protein tyrosine kinase to transform progenitor cell-derived smooth muscle cells to a premetastatic phenotype (Atay et al., 2014). Another report indicated that the Colorectal cancer cells with high invasive potential were detected to be significantly dependent on the concentration of exosomes including the signaling competent epidermal growth factor receptor (EGFR) ligand, inferring that exosome-mediated ligand shuttle causes cancer invasiveness and metastasis (Higginbotham et al., 2011). Exosome-modulated transferring of microRNA-221/222 from mesenchymal stem cells (MSCS) to gastric tumor cells significantly promotes migration and metastasis of these tumoral cells (Wang et al., 2014b).

Figure 3 Exosomes drive pre-metastatic niche formation.

The formation of the pre-metastatic niche is required for organ-specific metastatic tropism. The exosomes can move to the distant location for increasing the formation of pre-metastatic niche. The complementation of angiogenesis and induction of stromal and epithelial cell differentiation can be associated with a pro-tumor environment. Tumor-derived exosomes provide a pre-metastatic niche, through the polarization of tissue macrophage, suppression of dendritic cell maturation, induction of CAF (cancer-associated fibroblasts) via differentiation of fibroblasts to myofibroblasts. This effect can be performed via the mediation of intercellular cross-talk and subsequent adjustment of both local and distant microenvironments in an autocrine and paracrine fashion.

Role of exosomes in tumor immune escape

The current researches represented that tumor-derived microvesicles may function as immunosuppressive effects. Exosome-mediated communication among cancer cells and the immune system is triggered recruiting pro-cancerogenic immune cells (Fig. 4). Also, tumor-derived exosomes are being utilized as an effective source of tumor antigen to induce dendritic cells (DCS), causing a shuttle of tumor antigens to DCs and including CD8+ T cell-related anti-tumor outcomes. The exosomal tumor-carried TGF-β1 deviated IL-2 modulates in favor of regulatory T cells and away from cytotoxic cells (Bu et al., 2011; Clayton et al., 2007). Also, tumor-derived exosomes can activate myeloid-derived suppressor cells (MDSC). The MDSCs by inhibiting the T cell reaction can apply immunosuppressive functions in cancer. tumor-derived microvesicles from several tumor cell lines modulate synthesis of interleukin-6 (IL-6) in MDSCs via the activation of Toll-like receptor 2 through the membrane-associated heat shock protein 72 (HSP 72). Making of IL-6 outcomes in an autocrine phosphorylation of stat3 in MDSCs can enhance their immunosuppressive function (Chalmin et al., 2010; Nagaraj & Gabrilovich, 2012). The miRNA shuttled via cancer cell-derived exosomes may function as ligands through attaching to the Toll-like receptors and activate the inflammation. Indeed, it was cleared that oncogenic miR-21 and miR-29a released from the exosomes derived from highly metastatic lung carcinoma cells can bind to the human and murine TLRs (Fabbri et al., 2012).

Figure 4 Regulation of immune responses by extracellular vesicles.

The tumor-derived microvesicles may function as immunosuppressive effects. Exosome-mediated communication among cancer cells and the immune system is triggered recruiting pro-cancerogenic immune cells. The regulation of immune response in a procedure of prevention tumor diagnosis and anti-tumoral immune functions through impairing the function of effector T cells and natural killer cells (NK cells) can induce mobilization of neutrophils, and differentiate T-helper cells toward a T-regulatory cell phenotype.

Role of exosomes in mediating Insensitivity to a drug in cancer

Exosomes via several mechanisms may play pivotal role in the progression of therapy resistance in cancer cells. Tumor-derived microvesicles can shuttle multi-drug resistance (MDR)-associated miRNAs and proteins to target cells. These illustrated that several major classes of anticancer drugs and their metabolites can be encapsulated and exported through exosomes outside of the cells, and shedding of these extracellular vesicles (EVs) is intimately associated with insensitivity to a drug (Fig. 5) in various human cancer cells (Corcoran et al., 2012; Safaei et al., 2005; Shedden et al., 2003; Wei et al., 2014). Recently, emerging evidence illustrated that miR-21 was shuttled from cancer-associated adipocytes of fibroblasts to the various tumor cells, where it can inhibit ovarian cancer apoptosis and induce the Paclitaxel resistance through binding to its new direct target, apoptotic protease activating factor-1 (APAF1) (Yeung et al., 2016). Besides, there is plenty of interest in insensitivity to a drug through exosome-mediated shuttle of miRNAs. Several studies suggested that breast cancer cells resistant to various drugs (Docetaxel-Adriamycin-Tamoxifen) may shuttle the resistance to sensitive cells in part via exosomal miRNA exchange (Chen et al., 2014). Moreover, PTEN is reduced in exosomes therefore applying biological acts in target cells. The loss of function of PTEN enhances resistance to sensitivity and chemotherapeutic of mTOR, which inhibits in breast cancer cells and, afterwards, PTEN exosomal shuttle, could be drew out as a shuttle mechanism or drug resistance changes (Steelman et al., 2008). Besides exosomes through regulating their binding to tumor cells may counteract the efficacy of antibody drugs. The exosomes-originated lymphoma carry CD20 can bind to the anti CD-20 antibody therapeutics and induce the preserving of target cells from antibody attack (Aung et al., 2011). Thus, the exosomes-derived cancer cells can be utilized as a procedure of cancer chemotherapy resistance of special cancer cells to characteristic drugs.

Figure 5 Exosomes as mediators of drug resistance.

Drug resistance applies for a critical role in various cancer treatments. There are different mechanisms of drug resistance even multi-drug resistance (MDR) such as drug efflux, triggered by extracellular vesicles, which can make the defeat of the whole remedy. The tumor-derived exosomes can induce tumor cells to promote drug resistance through sending out the tumor drugs or inhibiting antibody-based drugs.

Use of exosomes as a tumor diagnostics and biomarkers

The indicating of significant functional roles of exosomes in approximately all aspects of tumor cells was preparing the opportunities for enhancement of these nanovesicles as a considerable diagnostic biomarkers and remedial targets. The exosomes derived-human tumor cells are enriched with mRNAs, proteins, and miRNAs which are more plentiful in tumor than in healthy noncancerous cells (Roma-Rodrigues, Fernandes & Baptista, 2014). One of the principal beneficiaries of the utilize of these nanovesicles as a valuable biomarkers is the feasibility of a fast pathology detection by minimally invasive procedures (Li & Bahassi, 2013). The existence of exosomes in the blood circulatory system and shedding these nanovesicles into biological fluids such as urine, saliva, and ascites of exosomes containing biomarkers in several subtypes of human tumor cells can be obtained the minimally invasive “liquid biopsies” (for example blood collection) for clinical use (Zhang & Grizzle, 2014). Also these microvesicles are really resistant under variant storage situations containing short-term storage at 4°C for 96 h or long-term storage at −70°C (Taylor & Gercel-Taylor, 2008). These quality attributes of the circulating serum exosomes can be used as a considerable biomarkers for early diagnostics of cancer cells and personalized cancer therapies. Several in vitro studies suggested that exosomes derived human tumor cells can be utilized as a remarkable biomarker to diagnose cancer cells through applying the methods of proteomics and transcriptomics (Aushev et al., 2013; Dijkstra et al., 2014). Also, the enhanced levels of exosomes in blood plasma specimens of colon carcinoma patients was considerably linked to the weakly differentiated tumor cells and the declined entirely survival (Silva et al., 2012). Another study illustrated that exosomal EDIL3 and fibronectin in circulating EVs can utilize as pivotal biomarkers of early stage breast cancer through applying ELISA methods (Moon et al., 2016). Recent report showed that PCA3 and TMPRSS2:ERG, two established proteins exist in urinary exosomes from prostate cancer proteins which are detected as a potential biomarkers through label-free liquid chromatography-tandem mass spectrometry (LC-MS/MS) (Nilsson et al., 2009). These finding show that the bodily fluids originated exosomes may be an important noninvasive marker for the early tumor detection.

Use of exosomes as a cancer therapeutic

Emerging reports indicate that various clinical researches illustrated the role of exosomes as cancer remedies, and a few main adverse effects were identified for applying of these nanovesicles in cancer therapy. Figure 6 indicates the remedies that were proposed for therapy of cancers based on exosomes characteristics.

Figure 6 The main groups of exosome-based therapies.

This overview includes impairing the secretion of exosomes via cancer cells and removing cancer derived exosomes, including bioactive molecules, from the blood (or other body fluids) of cancer patients; using exosomes, naturally-equipped nanocarriers, including microRNA (miRNA), small interference RNA (siRNA), and/or anticancer drugs for targeting delivery to tumor cells; the exosomes molecular composition indicates their cells of origin, may confer special cell or tissue tropism; applying exosomes as potent cell-free peptide-based vaccine demonstrate an remarkable strategy to inhibit tumor development; exosomal miRNAs can contribute to exosome-mediated cell–cell communication and induce anticancer features.

(1) Trough secretion of exosomes, tumoral cells trigger the alteration of the local and systemic tumor environment to induce tumor growth promotion, metastasis and insensitivity to drugs. Thus, either the destruction of exosome-dissemination pathway through tumor cells or the removal of these nanovesicles from the blood circulatory system may create an effective method for cancer therapy. The Tinzaparin (a low-molecular-weight heparin) can trigger tissue factor pathway inhibitor (TFPI) secretion from cancer cells, and also the recombinant TFPI induce suppression of tumor-derived exosomes causing migration of tumor cells (Gamperl et al., 2016). Lately for the elimination of extracellular vesicles (EVs) from the blood circulatory system, a therapeutic hemofiltration process which is called ADAPT™ (adaptive dialysis-like affinity platform technology) is applied. Whenever the patient’s blood plasma samples transfer via ADAPT™ system, plasma specimens factors by the porous fibers are interacted with the immobilized affinity agents to that target molecules are particularly absorbed while unbound serum factors and blood cells can pass through this system (Logozzi et al., 2009).

(2) As a delivery system, exosomes are considerably utilized as vehicles loaded with multiple anticancer drugs, siRNAs, and miRNAs for several cancer therapeutic cargos. The lipid bilayer membrane of these nanovesicles forms a natural protective shelter, thus enhances the cellular internalization of the encapsulated anti-cancer drugs. Regarding to the exosomes originated from autologous cancer cells, these nanovesicles can cause minimal toxicity when being shuttled into the target cells and can be less immunogenic than artificial delivery vehicles. Also their naturally small size can permit them to elude phagocytosis through the mononuclear phagocyte system (MPS) and simplify their extravasation via tumor blood vessels and their subsequent release in target cancer tissues. The various researches illustrated that prosperous delivery and tumor inhibition utilizing this procedure. The enhancement of colorectal and breast xenograft cancers in vivo by applying the Doxorubicin loaded into exosomes or using exosome-mimetic nanovesicles can be suppressed. Thus, the efficacy of Doxorubicin was widely promoted through targeting the immature dendritic cell exosomes into cancer tissues (Jang et al., 2013; Tian et al., 2014). Also the another greatly utilized antimitotic chemotherapeutic drug is Paclitaxel that can be loaded into microvesicles through sonication, and these loaded microvesicles have 50 times more cytotoxicity than free Paclitaxel for drug resistance tumor cells in vitro. Besides the exosome-encapsulated Paclitaxel can considerably block murine Lewis lung cancer pulmonary metastases and decline size of tumor in the mouse model (Kim et al., 2016).

(3) Exosomes can be utilized to target special tissues or organs because of the having special cell tropism according to their traits. Applying the well-characterized exosomal membrane protein (Lamp2b) for expressing the targeting peptide instantly below the signal peptide sequence including the targeting peptides RVG and iRGD were prosperously inserted within these nanovesicles from immature dendritic cells to target either brain or cancer tissues. This method considerably promoted the cellular uptake of the nanovesicles in the tissue of interest, and enhanced the specificity of the remedy, and also reduced the toxicity of drugs delivered through exosomes (Jang et al., 2013; Tian et al., 2014).

(4) In addition to anticancer therapeutic drugs, exosomes can likewise deliver several tumor antigens, nanobodies, apoptotic-containing proteins, proteasomes, deficient or mutant anti-apoptosis proteins, tumor and tissue-specific peptides, transferrins, and lactoferrins within tumor cells for targeting remedy (Aspe et al., 2014; Cho et al., 2005; Hall et al., 2016; Hung & Leonard, 2015; Kooijmans et al., 2016; Lai et al., 2012; Malhotra et al., 2016).

(5) Also, exosomes based cell-free vaccines could indicate an alternating to dendritic cells (DCs) treatment for inhibiting tumor development through the function of exosomes in the immune system. So researchers detected that the nanovesicles originated from peptide-pulsed DCs, can present antigens to T cells to affect their immune response. These DC-originated exosomes include MHC-peptide complexes and co-stimulatory molecules on their membrane, that permit them to continue antigen presentation and increase immunization in mice comparing with antigen-presenting DCs (Luketic et al., 2007).

(6) Moreover, miRNAs are widely detected in exosomes derived cancer cells or isolated from bodily fluids that contribute to exosome-mediated cell–cell communication, and induce anti-cancer traits. For instance EGFR-specific binding peptide GE11 can lead let-7a-containing exosomes to EGFR-positive cancer cells, that considerably suppressed EGFR-positive human breast tumor cell development in a heterograft mouse model (Ohno et al., 2013; Yu et al., 2015).

Functions and Remedial Roles of Exosomes in different kinds of Human Cancers

The tumor exosomes originated from the ascites of a very aggressive murine T-cell lymphoma (EVs A) can effect on dendritic cells activity, thus disturbing the immune system to distinguish and destroy cancer cells. Also, the expression of marker-proteins including ALIX, TSG-101, CD63, CD81, and CD9 has detected in EVs A. This research illustrated that EVs A triggered both humoral and cellular immune reactions. Altogether, the outcomes indicated that the endosome-originated EVs secreted via an advanced-stage T-cell lymphoma, stimulated a special immune reaction (Menay et al., 2017). Besides, exosomes released through chronic myelogenous leukemia (CML) cells remedied with Curcumin, which originated from the plant Curcuma longa, has the anticancer effects, include a wide quantity of miR-21 that is transported into the endothelial cells in a biologically active form. The treatment of HUVECs with CML Curcu-exosomes diminished RhoB expression and conversely modified endothelial cells motility. The research illustrated that the addition of CML control exosomes to HUVECs induced promotion of IL8 and VCAM1 levels, but Curcu-exosomes returned this efficacy, therefore, diminished their angiogenic properties. Overall, this research showed that besides Curcumin reduces the exosome’s capability to enhance the angiogenic phenotype and to modify the endothelial barrier organization (Taverna et al., 2016). As exosomes emerge as a novel manner of intercellular communication, the cargo includes through exosome is formed via somatic evolution. Regarding evaluating the effect of exosomes originated from several melanoma-related cell lines on primary CD8+ T cells act, exosomes from each of the cell lines were different. The B16F0 exosomes dose-associated inhibited T-cell proliferation. Notwithstanding, Cloudman S91 exosomes enhanced T-cell proliferation and Melan-A exosomes load an insufficient impact on primary CD8+ T cells. Importantly, B16F0 exosomes suppressed T-cell proliferation through high-expressed of PTPN11 to tumor permeating lymphocytes would escape the extracellular control of the immune checkpoints (Wu et al., 2015). Regarding increasing evidence, extracellular vesicles (EVs) are inherently trigger intercellular relation through shuttling molecular information between cells. Therefore, the autologous cancer-cell originated EVs can be utilized as helpful carries of Paclitaxel to the prostate cancer cells, bringing the drug into the cells via an endocytic process and released into the cell cytosol leading to cell death. Most considerably, the EV-mediated delivery promoted the cytotoxic efficacy of the drug. This research suggested that the autologous EVs may be helpful for impressive transporting of chemotherapeutic agents to prostate cancer cells (Saari et al., 2015). It is noteworthy that the effective role of exosomes in relevance among cancer cells and surrounding stroma indicated that the TrkB expression in exosomes is necessitated including aggressiveness phenotype. In this report, the YKL-40 silencing contributes for reducing of TrkB, sortilin and P75NTR expression, related to a low aggressive phenotype. The release of TrkB in exosomes from normal glioma cells was able to relieve both migration and activation of YKL-40-inactivated cells. Furthermore, TrkB-containing exosomes may be remarked as a considerable biomarker for glioblastoma diagnosis (Pinet et al., 2016). Also, MVs originated from HLSC (MV-HLSC) can suppress the growth of hepatoma tumors through shuttling genetic information that mediates with deregulated survival and proliferation of these cells. The antitumor effect of MV-HLSC was relevant to the decreased internalization, due to the lack of CD29 on MV-fibroblast or a decreased expression of antitumor miRNAs including miR-24, miR-31, miR-122, miR-125b, miR-223, and miR-451. Consequently, the promoted internalization was not relevant to an increased biological activity when MV-fibroblast expressing CD-29 were utilized. Hence, the various composition of miRNA content between MV-HLSC and MV-fibroblast was the remarkable reason for the various biological actions. Therefore, the transferring of these miRNAs through MVs originated from stem cells may suppress tumor growth development and stimulate apoptosis (Fonsato et al., 2012). Another research illustrated that exosomes originated from curcumin-pretreated H1299 cells were utilized to remedy BEAS-2B cells, which triggered proliferation, colony organization, and migration of BEAS-2B cells. Curcumin is a new drug lung cancer remedy. Although, the procedure associated with the antitumor effect of curcumin is related to the promoted expression of TCF-21, triggered through a low expression of DNMT1. Therefore, mechanism of curcumin is remarkable in cancer remedy, and creates the pivotal biomarkers for developing cancer diagnostic and remedial procedures (Wu et al., 2016). The new research suggested that PSC (Pancreatic Stellate Cells) originated exosomes can trigger and elevate the proliferation and migration of PANC-1 and SUIT-2. The exosomes differentially varied expression of a plethora of genes controlling multiple cellular procedures containing cell cycle, cellular assembly and organization, DNA replication, recombinant and repair, cell death and survival, cellular development, and growth in the recipient cancer cells. Also, three chemokines, CCL20, CXCL1, and CXCL2 were detected high expressed in exosome-treated cancer cells. Besides, GPC1 (glypican-1), a glycoprotein discovered in PSC exosomes, as a pivotal biomarker to distinguish PDAC (Pancreatic Ductal Adenocarcinoma), and as a tumor promoter shuttled among cells through exosomes (Ali et al., 2015; Charrier et al., 2014; Farrow et al., 2003). The exosome/staphylococcal enterotoxin B is a considerable sample for apopto-immunotherapy. The contribution of Exo and its lipid rafts in this structure assigns the feasibility of binding to pancreatic cancer cells. SEB and the characterized lipid rafts trigger the apoptotic signal both through extrinsic and mitochondria-dependent pathways. Also, the presence of tumor antigens associated with superantigen causing promotion of specific antitumor immune response (Mahmoodzadeh et al., 2014). Even more recent research detected that gastric cancer cells may release exosomes for transferring apoptotic signals without direct cell–cell contact to anti-cancer T cells. The Cbl-b and Cbl-c of ubiquitin ligases might have a considerable role in exosome-induced apoptosis of Jurkat T cells through enhancing PI3K proteasome degradation, that can cause inactivation of PI3K/Akt signaling, therefore led to activation of caspase 3, 8, and 9. Thus, relation among exosomes and immune response is presumably to assign considerable point of view through the process of tumor immune inhibition (Qu et al., 2009). Recently study illustrated that miR-375 promoted the growth inhibitory effect, Cell progression and dissemination of colon cancer through the Bcl-2 pathway. Therefore the miR-375 down-regulated in metastatic CRC, and it has important role for Bcl-2 blocking, with the significant minimally invasive prognostic biomarker for CRC through suppression of malignant proliferation and dissemination (Zaharie et al., 2015). Further, research reported that a HIV-Nef SMR-originated peptide suppressed the progress of human breast tumor cells through arresting cancer cell cycle and including blockade of exosome secretion. The SMR peptide inhibited the cancer cell cycle through G2/M phase boundary. While the SMR peptide and chemotherapeutic drugs were compound to remedy cancer cells, PEG-SMRwt-Clu synergically enhanced the anti-proliferative efficacies of drugs, considerably promoted the tumor cell growth suppression efficacy of drugs and inhibited exosomes secretion in breast cancer cell lines MCF-7 and MDA-MB-231 cells. Therefore the considerable usage of PEG-SMRwt-Clu peptide is pivotal process for the prevention and therapy of human breast tumor cells (Huang et al., 2017).

Conclusions

Prosperity in remedy against intricate cancers relies on our full comprehension of the complications among various components within tumors. The above studies supported the viewpoint that exosomes can play a pivotal role in the growth, and promotion of cancer cells via regulation of intercellular communication into the tumor microenvironment through the release of several biological molecules ranging from virions of mRNA, miRNA, protein, lipid, and DNA cargos. An overview of the roles of exosomes in different types of cancer and the molecular mechanisms that have been used to evaluate the effect of EV in cancer progression and metastasis were presented in Table 1. The exploration of exosomes derived cancer cells contents may permit the progression of new diagnostic and remedial procedures, with minimally invasive approaches. Exosomes derived cancer cells also can cause cancer cell development and metastasis through suppressing the immune response and via enhancing chemoresistance by elimination of chemotherapeutic anti-cancer drugs. So they might be significant targets for remedial interventions through their alternation or elimination. The field of nanotechnology has widely benefitted exosome research to load nanovesicles with tiny molecules or drugs for cancer treatment because of their small size, lack of toxicity and target specificity toward prosperous immunotherapy in clinical trials. Besides exosomes can be considerable biomarkers for diagnosis of cancer and targeted remedy due to their nearly display the situation of their parental cells that are relatively constant in the blood circulation and could be feasibility obtained from body fluids. Most significantly the role of miRNA in the context of exosomes for targeting inactivation of cancer-causing miRNAs would probably provide a novel strategy for most cancers. It is expected that further research on these microvesicles will not only determine great potential and hopeful effect on their functions in the pathogenesis of cancer but also will open the new strategies for cancer diagnosis and remedies.

Table 1 Overview of the role of exosomes in multiple kinds of cancer.

Exosomal cargos	Cancer cell types	Methods	Clinical values	References	
ECM1, APN, APOC4, and AZGP1	Serum samples were collected from normal, and healthy women, women with NTMnb, and women with BCa-NTMnb
The HMLE and SUM149 cell lines	FACS analysis, Western blotting and, Immunofluorescence analyses	APOC4, APN, and AZPG1 as additive factors might possibly increase NTM (Bronchiectasis and nontuberculous mycobacterial disease) susceptibility via the modulation of immune function and triggering lipolysis.	Philley et al. (2017)	
PEG-SMRwt-CLU peptide	The human breast cancer cell lines MCF-7 cell line, a noninvasive estrogen receptor positive (ER+), and MDA-MB-231 cell line ER negative
The MCF-10A cell line, a non-tumorigenic epithelial cell line	Exosomes characterization by acetylcholinesterase (AchE) assay, Exosome nanoparticles tracking analysis (NTA), and Western blotting	The SMR peptide inhibited breast cancer cell growth, reduced exosome secretion without increasing the cytotoxic effects of chemotherapy or promoting apoptosis.	Huang et al. (2017)	
MiR-10b	The human breast cancer cell lines MCF-7 and MDA-MB-231 cells
The human mammary epithelial cell lines MCF-10A, and HMLE cells
The human embryonic kidney cell line HEK-293T cells	qRT-PCR analysis, and Western blotting	MiR-10b as an exosomal miRNA that elevated cell invasion in HMLE cells through targeting HOXD10 and KLF4, indicating the invasive tumor cells may utilize exosomal miRNAs as a means for their advance.	Singh et al. (2014)	
MiR-198, MiR-26a, MiR-34a, MiR-49a, let-7a, MiR-328,
MiR-130a, MiR-149, MiR-602, MiR-92b	The human breast cancer cell lines, culture supernatants from MCF7, and MDA-MB231 cells	qRT-PCR analysis, and Western blotting	The extracellular vesicles carry oncogenic proteins and miRNAs, which may further be applicable for early detection of breast malignancy as well as delineating the possible role of extracellular vesicles in tumorigenesis and metastasis.	Kruger et al. (2014)	
C6 Ceramide	The human breast cancer cell line MDA-MB 231 cells	qRT-PCR analysis, and Immunocytochemistry assay	Exogenous C6 ceramide, a sphingolipid known to induce exosome secretion, also induced secretion of BCRP-associated exosomes, while siRNA-mediated knockdown or GW4869-mediated inhibition of neutral sphingomyelinase 2 (nSMase2), an enzyme generating ceramide, restored cellular BCRP.	Kong et al. (2015)	
HCV RNA (exoRNA)	The human breast cancer cell lines (IRDs Responders) 1833, MDA-MB-231, Hs578T, MDA-MB-436, MDA-MB-157, and HCC1937
The human breast cancer cell lines (IRDs non-Responders) SKBR3, T47D, MCF-7, HCC70, and MDA-MB-468	Chromatin immunoprecipitation and primary transcript analysis, and Mammosphere analysis	Stromal cells orchestrate an intricate cross-talk with BrCa cells by utilizing exosomes to instigate anti-viral signaling. This expands BrCa subpopulations adept at resisting therapy and re-initiating tumor growth.	Boelens et al. (2014)	
Hsp70 (an exosomal protein marker)	The epithelial like breast cancer cell line MDA MB-231 cells	qRT-PCR analysis, and Western blotting	The EXO/SEB, two immune inducer substances, was able to induce cytostatic events through apoptosis in insensitive human ER—breast cell line. The EXO/SEB considerably decreased the cell proliferation and stimulated apoptosis via increasing the expression level of Bak, and Bax, and raised the activity of caspase-3 and caspase-9.	Hosseini et al. (2014)	
RPL27A, GDF11, EPS15L1, NUDT16, TRAK2, CCDC11, BEND6, ZNF114, IFNAR1, PITPNM3, ENSA, ALKBH7, APLP2, VAPA, SNRPB, SAR1B, DCAF16, FAM134B, GJC1, and
MSLN	The human metastatic mammary gland epithelial adenocarcinoma cell line MDA-MB-231, and human submandibular gland (HSG) cells	Western blot analysis	The breast cancer-derived exosome-like microvesicles are capable of interacting with salivary gland cells, altering the composition of their secreted exosome-like microvesicles.	Lau & Wong (2012)	
OIP2	The human breast cancer cell lines MDA-MB-231 cells, and MCF-7 cells	qRT-PCR analysis, and Enzyme-linked Immunosorbent Assay (ELISA)	Monad-mediated degradation is one of the mechanisms that determines the stability of amphiregulin mRNA and that Monad-amphiregulin axis plays an essential role in the invasion of breast cancer cells.	Saeki et al. (2013)	
Wnt10b	The immortalized WT mouse embryonic fibroblasts (MEFs) and the p85α − ∕ −
MEFs
The human breast cancer cell line MDA-MB-231
The mouse breast cancer cell line 4T1	qRT-PCR analysis, and Western blotting	Paracrine Wnt10b from p85 α-deficient fibroblasts can promote cancer progression via EMT induced by the canonical Wnt pathway. Moreover, exosomes have a key role in paracrine Wnt10b transport from fibroblasts to breast cancer epithelial cells. Thus p85 α expression in stromal fibroblasts has a pivotal role in regulating breast cancer tumorigenesis and progression.	Chen et al. (2017)	
ERG, PCA3, and SPDEF	The urine samples of prostate cancer (PCA)-free men 50 years or older
The urine samples of Men with a history of invasive treatment for benign prostatic disease	qRT-PCR analysis	The ExoDx Prostate IntelliScore is a validated, easy to administer, noninvasive urine exosome gene expression assay gene signature derived from genes known to play a pivotal role in prostate cancer initiation and development including ERG, PCA3, and SPDEF, with the potential to decrease the total number of biopsies performed in men with a suspicion of prostate cancer.	McKiernan et al. (2016)	
Paclitaxel (PtX), a widely used antimitotic cancer therapeutic	The human prostate cancer cell lines LNCaP and PC-3 PCa cells	Nanoparticle tracking analysis (NTA), and Western blotting	Cancer cell-derived EVs can be utilized as beneficial carriers of Paclitaxel to their parental cells, bringing the drug into the cells via an endocytic pathway and promoting its cytotoxicity. Thus, autologous EVs may have potential for effective delivery of chemotherapeutics to cancer cells.	Saari et al. (2015)	
Claudin 3 (CLDN3)	The human metastatic PC3 and benign PNT1A prostate cell lines
The blood plasma of patients with prostate cancer	Immunoblotting, Enzyme-linked immunosorbent assay (ELISA), and Western blotting	CLDN3 is an exemplary exosome-based circulating biomarker which candidate for prostate cancer from in vitro profiling of cancer exosomes over in silico identification and in vitro retesting to clinical validation. Besides, CLDN3 plasma levels were considerably increased in patients with high Gleason score, pointing to a potential predictive value of this marker.	Worst et al. (2017)	
B7-H3 (CD276)	The human prostate cancer cell lines (androgen-responsive: LNCaP, 22RV1 and -irresponsive: DU145)
Normal human dermal fibroblasts (NHDF)	Western Blot Analysis	The release of exosome-like microvesicles can promote during proliferative senescence in normal human diploid fibroblasts. Moreover, these exosomes were enriched in B7-H3 protein, a recently identified diagnostic marker for prostate cancer and an abundance of exosomal shuttle RNA.	Lehmann et al. (2008)	
The immunomodulatory cytokine IL-6, and the
pro-angiogenic factors IL-8, VEGF, and MMP2	The malignant melanoma cell lines,
Mewo, SKmel28, A2058, A375, and HTB63 (HT-144)
MS1 murine endothelial cells	qRT-PCR analysis, Enzyme-linked immunosorbent assay (ELISA), and Western blotting	The non-canonical Wnt protein WNT5A signaling induces a Ca2+-dependent release of exosomes containing the immunomodulatory and pro angiogenic proteins IL-6, IL-8, VEGF, and MMP2 in melanoma cells.	Ekström et al. (2014)	
Histones (H2A, H2B, H3.1 and H4), heat shock proteins
(GRP78 and HSC71), and the tetraspanin CD81	The C57BL/6 derived melanoma cell lines B16-F1, and B16-OVA (B16-F0 cell line
The C57BL/6 derived thymoma derived EL4
cell line	Flow cytometric analyses, and Western blotting	Extracellular vesicles (EVs) have been implicated in thrombotic events (the second highest cause of death in cancer patients) and tumor vesicles contribute to the anti-cancer immune response.	Muhsin-Sharafaldine et al. (2016)	
_	Metastatic melanoma cell lines Me 30966	Flow cytometric analyses, and Western blotting	The enhanced drug delivery time of Exo-AO to melanoma cells as compared to the free AO, improving the cytotoxicity of AO. Thus, Exo-AO has a great potential for a real exploitation as a novel theranostic approach against tumors based on AO delivered through the exosomes.	Iessi et al. (2017)	
CD9, CD63, CD81, Cluster 1 (MiR-216a, MiR-217, MiR-129-5p, and MiR-203), Cluster 2 (MiR-9, MiR-125a-5p, MiR-25, MiR-125b, MiR-335, and MiR-19a), Cluster 3 (MiR-370, MiR-210, MiR-320a, MiR-124, MiR-107, and MiR-486-5p)	The blood plasma samples of patients with isolated liver metastases from uveal melanoma
The human malignant melanoma cell lines A375, and MML-1
The human breast cancer cell line, HTB-133
The human lung carcinoma cell line, HTB-177
The human mast cell line, HMC-1.2	Flow cytometry assay, qRT-PCR analysis, and Western blotting	Melanoma exosomes are released into the liver circulation in metastatic uveal melanoma, and is associated with higher concentrations of exosomes in the systemic circulation. The exosomes isolated directly from liver circulation contain miRNA clusters that are different from exosomes from other cellular sources.	Eldh et al. (2014)	
MAGE A3 (168–176)/class I, MAGE A3 (247–258)/class II, tetanus toxoid/class II, MAGE A3 (168–176)/class I, MAGE A3 (247–258)/class II, MAGE A3 (168–176)/class I, MAGE A3 (247–258)/class II, tetanus toxoid/class II, MAGE A3 (168–176)/class I, MAGE A3 (247–258)/class II	Fifteen patients bearing melanoma (stage IIIB and IV, HLA-A1+, or -B35+ and HLA-DPO4+ leukocyte phenotype, tumor expressing MAGE3 antigen)	Flow cytometry assay, qRT-PCR analysis, and Enzyme-linked immunosorbent assay (ELISA)	The case report of MART1 antigen spreading and MHC class I loss variant suggested that exosomes mediated bioactivity in vivo, supporting to conduct Phase II clinical trials. Thus, the first exosome Phase I trial highlighted the possibility of large scale exosome production and the safety of exosome administration.	Escudier et al. (2005)	
Housekeeping proteins (CD63 and Rab-5b) and a tumor-associated marker (caveolin-1)	The human metastatic melanoma cell lines Me501, and MeBS cells
The osteosarcoma (SaOS-2) and colon carcinoma cell lines
The blood plasma samples of melanoma patients	Flow cytometry assay, Enzyme-linked immunosorbent assay (ELISA), and Western blotting	Plasma exosomes expressing CD63 or caveolin-1 were significantly promoted in melanoma patients as compared to healthy donors. Moreover, caveolin-1+ plasma exosomes were remarkably increased with respect to CD63+ exosomes in the patients group.	Logozzi et al. (2009)	
MiR-21, MiR-34 a, and MiR-146a	The blood serum sampling of Uveal melanoma (UM) patients and healthy donors	Flow cytometry assay, and qRT-PCR analysis	MiRNAs differentially expressed in UM patients comparing with healthy donors. Most alterations were common to vitreous humor (VH), and vitreal exosomes (upregulation of miR-21,-34 a,-146a). Interestingly, miR-146a, miR-34a, and miR-146a were upregulated in the serum of UM patients, as well as in serum exosomes.	Ragusa et al. (2015)	
Tyrosinase related
protein-2 (TYRP2), very late antigen 4 (VLA-4), heat shock protein 70 (HSP70), an
HSP90 isoform, and MET oncoprotein	The human peripheral blood samples of melanoma patients
8–10 week-old C57Bl/6 female mice
The human breast cancer cell lines MCF-7, SkBr3, and MDA-MB-231
The cellosaurus cell line AsPC-1
The Lewis Lung carcinoma cell line LLC
The colon carcinoma cell lines SW480, and SW620
The human melanoma cell lines B16-F10, and B16-F1	Flow cytometry assay, qRT-PCR analysis, and Western blotting	Decreasing Met expression in exosomes reduced the pro-metastatic behavior of BM cells. Interestingly, MET expression was increased in circulating CD45−C-KITlow/+TIE2+ BM progenitors from metastatic melanoma subjects. RAB1a, RAB5b, RAB7, and RAB27a were highly expressed in melanoma cells and Rab27a RNA interference diminished exosome production, preventing BM education, tumor growth and metastasis.	Peinado et al. (2012)	
Superparamagnetic iron oxide nanoparticles 5 (SPION5)	The C57BL/6 mouse model
The mouse B16-F10 (CRL 6475) melanoma cells	MRI analysis	The melanoma exosomes appear to be trafficking to a particular microanatomical destination in lymph nodes known as the subcapsular sinus. Thus, SPION5 loaded exosomes might be particularly tailored through endogenous molecular cell based nanofactories and/or exogenous synthetic exosome modification to simultaneously detect and treat pathogenic microenvironments.	Peinado et al. (2016)	
Stabilin 1 (MS-1), Ephrin R β4, Integrin αvβ3, MAPK 14, urokinase plasminogen activator (uPA), laminin 5, collagen 18, G-α13, VEGF-B, Increased hypoxia inducible factor 1α (HIF1-α), thrombospondin 1 (Thbs1), Tumor microenvironment associated tumor necrosis factor α (TNF-α)	The mouse B16-F10 (CRL 6475) melanoma cells
Male 6- to 8-week old albino C57/BL6 mice	qRT-PCR analysis	Melanoma exosomes are capable of directly tuning a remote lymph node toward a microenvironment that facilitate melanoma growth and metastasis in lymph nodes even in the local absence of tumor cells.	Hood, San & Wickline (2011)	
PTPN11	Eight- to 12-week-old transgenic B6.Cg-Thy1a/Cy Tg (TcraTcrb)8Rest/J female mice
The murine melanoma cell lines B16F0, Cloudman S91 (clone M-3), and CTLL-2 cells
An immortalized mouse melanocyte cell line, Melan-A cells
The TH1 cell model, 2D6 cells	Flow cytometry assay, qRT-PCR analysis, and Western blotting	The tumor-derived exosomes can upregulate PTPN11, which is a phosphatase involved in immune checkpoint pathways, to suppress T cell proliferation and are sized to accumulate within the tumor microenvironment.	Wu et al. (2017)	
Fatty acid oxidation (FAO)	The murine 3T3-F442A preadipocyte line
Eight week old C57BL/6J male mice
The human adipose tissue samples
The human melanoma cell line SK-MEL-28
The human metastatic melanoma cell line 1205lu	Nano-LC MS/MS analysis, and Western blotting	The adipocyte exosomes stimulate melanoma cell migration and invasion. These exosomes, particularly enriched in proteins implicated in fatty acid oxidation (FAO), induce metabolic reprogramming in tumor cells in favor of FAO, enhancing aggressiveness.	Lazar et al. (2016)	
–	The B16-BL6 murine melanoma cell line
Five-week-old male C57BL/6 and BALB/c mice	qRT-PCR analysis, and Dynamic light scattering, zeta potential assay	Through designing a fusion protein consisting of Gaussia luciferase and a truncated lactadherin, gLuc-lactadherin, and constructing a plasmid expressing the fusion protein, sequential in vivo imaging indicated that the B16-BL6 exosome-derived signals distributed first to the liver and then to the lungs which is helpful for tracing exosomes in vivo and that B16-BL6 exosomes.	Takahashi et al. (2013)	
LAMP-1, and CD9	The female C57BL/6 mice
The highly lung metastatic OVA expressing B16 melanoma cell line BL6–10OV A
The naive CD8+ T cells and ovalbumin (OVA)-pulsed splenic dendritic
cells (DCOV A)	Flow cytometry assay, Enzyme-linked immunosorbent assay (ELISA), and Western blotting	The natural CD8+25+ Tr cell-secreted EXOs are capable of suppressing in vivo DC-induced CTL responses and antitumor immunity, indicating that CD8+25+ Tr-released exosomal molecules may play a pivotal role in Tr cell-mediated immune suppression. CD4+25+ Tr cell suppression has been found to be related with cell–surface inhibitory LAG-3, Gal-1, Nrp-1 and TIGI molecules.	Xie et al. (2013)	
–	The C57BL/6 female mice and CD45.2+ OT-I transgenic female mice (8 to 12 weeks of age)
The CD45.1+ C57BL/6-Ly5.1 female mice (8 to 12 weeks of age)	Flow cytometry assay, Enzyme-linked immunosorbent assay (ELISA), and Western blotting	The therapeutic vaccination targeted to the tumor-draining lymph nodes (tdLNs) of B16F10 melanoma-bearing mice with Dexo released by DCs co-cultured with oxidized necrotic B16F10 cells as source of melanoma antigens and matured with poly (I:C) (Dexo (B16 + pIC)) raised both melanoma-specific effector CD8+ T cells in the tdLNs, spleen and tumor mass and tumor-infiltrating NK and NK-T cells, significantly reducing tumor growth and increasing the survival rate of diseased mice.	Damo et al. (2015)	
cisplatin (CisPt)	The human breast cancer cell line MCF7
The human metastatic melanoma cell lines Me30966and Me501
The human colon carcinoma cell line SW480
The Human PBMC (Peripheral Blood
Mononuclear Cells)
Female CB.17 SCID/SCID mice aged 4–5 weeks	Enzyme-linked immunosorbent assay (ELISA)	CisPt uptake by human tumor cells was markedly impaired by low pH conditions. Moreover, exosomes purified from supernatants of these cell cultures contained various amounts of CisPt, which correlated to the pH conditions of the culture medium.	Federici et al. (2014)	
MiR-21	The imatinib-sensitive CML cell lines K562-s, and LAMA84-s
The human Umbilical Vein Endothelial Cells (HUVEC)	qRT-PCR analysis, Flow cytometry assay, Enzyme-linked immunosorbent assay (ELISA), and Western blotting	The exosomes released by chronic myelogenous leukemia (CML) cells after Curcumin remedy deeply changed their molecular composition, acquiring antiangiogenic properties. Curcu-exosomes were enriched in miR-21 which was then shuttled in endothelial cells as a biologically active form.	Taverna et al. (2016)	
ALIX/PDCD6IP, TSG101,HLA-DR, RAB5A, CD63, CD81, MiR-21, MiR-155, MiR-146a, MiR-148a, and let-7g, human leukocyte antigen (HLA)-DR molecules, B cell-specific markers (CD19 and CD20) and tetraspanins (CD37, CD53, and CD82)	The blood plasma of patients with chronic lymphocytic leukemia (CLL)
The eight-week-old NSG mice
The Bone Marrow-Derived Mesenchymal Stem Cells (BM-MSCs)
The BM-derived stromal cell line HS-5, and the endothelial cell line HMEC-1
The primary PKH67-labeled CLL cells	Flow cytometry assay, qRT-PCR analysis, and Western blotting	Exosome uptake by endothelial cells promoted angiogenesis ex vivo and in vivo, and coinjection of CLL-derived exosomes and CLL cells enhanced tumor growth in immunodeficient mice. Also, the results showed a-smooth actin–positive stromal cells in lymph nodes of CLL patients.	Paggetti et al. (2015)	
MiR-1908, and MiR-298	The human myeloid leukemia (CML) cell line K562	qRT-PCR analysis, and Western blotting	The expression level of miRNAs were different among K562 cells and K562 cell-derived exosomes. Thus, selectively expressed miRNAs in exosomes may promote the development of CML via effects on interactions (e.g., adhesion) of CML cells with their microenvironment.	Feng et al. (2013)	
CD81, Alix, Tsg101, and TGF-β1	The human chronic myeloid leukemia cell line LAMA84 cells
The four-to-five week old NOD/SCID mice	qRT-PCR analysis, Enzyme-linked immunosorbent assay (ELISA), and Western blotting	The exosome-treated LAMA84 cells is associated with the reduction of BAD and BAX proteins, as well as an increase in the protein levels of BCL-xl, BCL-w. Moreover, CML exosomes stimulate the proliferation and survival of the producer cells via the activation of ERK, Akt and NF-kB.	Raimondo et al. (2015)	
VEGF, Tax, CXCR4, Nanog, MMP-9, N-Cadherin, α-SMA, MiR-21, and MiR-155	The leukemic cell lines HTLV-I negative (Molt-4) or positive (C81 and HuT-102)
The peripheral blood plasma of acute ATL patients
The human mesenchymal stem cell line MSCs	qRT-PCR analysis, Enzyme-linked immunosorbent assay (ELISA), and Western blotting	The cargo of HuT-102-derived exosomes included of miR-21, miR-155 and vascular endothelial growth factor. Also, HuT-102-derived exosomes not only deliver Tax to recipient MSCs, but also induce NF-κB activation leading to an alteration in cellular morphology, promote in proliferation and the induction of gene expression of migration and angiogenic markers.	El-Saghir et al. (2016)	
TGFβ1, latency-associated protein (LAP), CD9, CD81, CD34, and CD 117	The blood plasma of acute myeloid leukemia (AML) patients at diagnosis, post-induction CT, during consolidation CT, in long-term remission, and from healthy volunteers	Flow cytometry assay, Enzyme-linked immunosorbent assay (ELISA), and Western blotting	The changes in total exosomal protein levels and the presence of various forms of transforming growth factor-beta1 (TGF-b1) carried by AML exosomes reflect effects of remedy and might serve as indicators of leukemic relapse in AML patients. Besides, AML exosomes carrying an active form of TGF-b1 induced down-regulation of NKG2D expression in normal natural killer (NK) cells.	Hong et al. (2014a)	
HSP70, and ABL	The DBA/2 mice (Dilute Brown Non-Agouti)
The chronic myeloid leukemia (CML) cell line K562
The mouse lymphocytic leukemia cells L1210
The DBA/2 mouse leukemia cell line
The Menogaril-resistant mouse leukemia P388 cells	Flow cytometry assay, and Western blotting	The EXOK562-pulsed DCs activate CTLs in vitro, which kill target cells more powerfully than CTLs induced by EXOK562 alone or by DCs pulsed with cell lysates. Moreover, LEXs induce antileukemic immunity and that LEX-pulsed DCs have the more potent antigen-specific antileukemic effects, because all mice injected with non-pulsed DCs developed tumors.	Yao et al. (2014)	
CD63, CD81, CD34, CD200, CD44, and CD105	The human CD34+ leukemic cell line
The blood plasma samples of newly-diagnosed AML patients and from healthy volunteers	Flow cytometry assay, Enzyme-linked immunosorbent assay (ELISA), and Western blotting	The blast-derived exosomes can be quantitatively ameliorated from AML patients’ plasma and that their molecular profile recapitulates that of the blasts. These isolated exosomes are biologically-active, trigger immune inhibition and might be helpful for AML diagnosis and prognosis.	Hong et al. (2014b)	
CD63, CD81, and Tsg101	The chronic myeloid leukemia (CML) cell line K562
The human umbilical vein endothelial cells (HUVEC)
Four-week-old BALB/c nude mice	Immunoblot analysis, Endothelial tube formation assay, XTT cell viability assay, and Matrigel plug assay	Exosomes released by K562 CML cells are internalized via endothelial cells during tubular differentiation on Matrigel and are shuttled to neighboring cells via the formation of nanotubular structures connecting the cells. Also, these exosomes stimulate tube formation in endothelial cells via Src activation. While both imatinib and Dasatinib reduced exosome release from K562 cells, only Dasatinib blocked exosome effect on endothelial cells.	Mineo et al. (2012)	
Interleukin-8 (IL-8)	The human vascular endothelial cells (HUVECs)
The human chronic myeloid leukemic cell line The human peripheral blood mononuclear cells (PBMC)
Four week old BALB/c nude mice	qRT-PCR analysis, Flow cytometry assay, Enzyme-linked immunosorbent assay (ELISA), Immunoprecipitation assay and Western blotting	LAMA84 CML cells are illustrated that addition of exosomes to human vascular endothelial cells (HUVEC) induces an increase of both ICAM-1 and VCAM-1 cell adhesion molecules and interleukin-8 expression. Also, the treatment with exosomes from CML cells caused an increase in endothelial cell motility accompanied by a loss of VE-cadherin and β-catenin from the endothelial cell surface.	Taverna et al. (2012)	
Alix, CD81, Tsg101, Interleukin 3 (IL3), and Lamp2b	The human embryonic kidney cell line HEK293T cells
The chronic myelogenous leukemia cell lines LAMA84, K562, and Imatinib resistant K562 cells
Four to five weeks old female NOD/ SCID mice	qRT-PCR analysis, Atomic Force Microscopy (AFM) assay, and Western blotting	The HEK293T cells was engineered to express the exosomal protein Lamp2b, fused to a fragment of Interleukin 3 (IL3). The modified exosomes, including IL3-Lamp2B, which loaded with Imatinib, are able to particularly target tumor cells in vivo, causing the decrease in tumor size. Thus, the modified exosomes are able to deliver functional BCR-ABL siRNA towards Imatinib-resistant CML cells.	Bellavia et al. (2017)	
GATA1, FOXP3, SHIP1, ID1, E2F1, CEBP-a and -b, Myc, and MEF2C, specifically, nucleophosmin 1 (NPM1), FLT3, CXCR4, MMP9, IGF-IR, Let-7a, MiR-9, MiR-99b, MiR-150, MiR-155, MiR-191, MiR-223, MiR-146a, and MiR-150	The acute myelogenous leukemia (AML) cell lines HEL, HL-60, Molm-14, and U937
The blood plasma of AML patients
Igf-1r knockout (R−) mouse embryonic fibroblasts and R− cells expressing human insulin-like growth factor (IGF)-IR cDNA (termed R+)	qRT-PCR analysis, Flow cytometry assay, and Western blotting	Profiling the mRNA content of these microvesicles indicated the presence of transcripts relevant to AML prognosis (FLT3-ITD, NPM1), treatment (FLT3-ITD, IGF-IR, CXCR4), and niche function (IGF-IR, CXCR4, MMP9). Also, both miR-150 and CXCR4 mRNA are present in AML exosomes, miR-150 is highly enriched therein, and exosome transfer to Ba/F3 progenitor cells was associated with a loss of CXCR4 surface expression and consequent reduce in cell migration toward SDF-1a.	Huan et al. (2013)	
Statistical analysis indicated that out of a total of 4,232 proteins
729 were considerably up-regulated in high AAI exosomes and 498 were up-regulated in low AAI exosomes	The blood plasma of AML patient	Immunocytochemistry assay, Flow cytometry analysis	The expression of apoptosis-regulating proteins (B-cell CLL/lymphoma 2 - BCL-2, Myeloid Cell Leukemia 1 -MCL-1, BCL-2 like 1 - BCL-X and BCL-2-associated X protein - BAX) in AML blasts at diagnosis is associated with disease-free survival. The intraindividual ex vivo apoptosis-related profiles of normal lymphocytes and AML blasts within the bone marrow of AML patients were increasingly correlated. Also, apoptosis-resistant primary AML blasts, as opposed to apoptosis-sensitive cells, were able to up-regulate BCL-2 expression in sensitive AML blasts in contact cultures.	Wojtuszkiewicz et al. (2016)	
_	The human p190BCR−ABLdriven ALL cells line (ALL3)
The chronic myeloid leukemia (CML) cell line K562, R10(-), Mo7, and CML CD34+ cells
Eight to ten weeks old female NOD/ SCID mice	[3H]-Thymidine incorporation assay, Enzyme-linked immunosorbent assay (ELISA), and Western blotting	The HD ALL3 cells are able to secret exosomes in large quantities and that they are capable of trigger the growth of the LD ALL3 cells without which they will not survive. Direct stimulation of non-growing LD ALL3 cells using purified exosomes shows that the ALL3 cells can also communicate with each other by means of exchange of exosomes independently of direct cell–cell contacts or diffusible soluble stimulatory factors secreted by HD ALL3 cells.	Patel, Darie & Clarkson (2016)	
TGF-β1, Hsc70, and NKG2D	The chronic myeloid leukemia (CML) cell line K562
The viable imatinib-resistant cells (K562RIMT)	Flow cytometry analysis, and Western blotting	The Dasatinib promotes cellular apoptosis via suppression of Akt/mTOR activities, and prevents exosomal release via downregulation of beclin-1 and Vps34 -dependent autophagic activity, containing distinct Dasatinib-induced mechanisms of apoptotic response and exosomes release in imatinib-resistant CML cells.	Liu et al. (2016)	
Interleukin-8 (IL 8)	The chronic myelogenous leukemia cell line LAMA84 cells
The bone marrow-derived stromal cell line HS5 cells
Four-to-five week old male NOD/SCID mice	qRT-PCR analysis, Enzyme-linked immunosorbent assay (ELISA), and Western blotting	Serum IL 8 levels enhanced in hematologic malignancies compared to healthy controls and promoted expression of IL 8 and its receptors has been indicated in cancer cells and stromal cells illustrating that IL 8 may modulate tumors microenvironment. Thus, LAMA84-derived exosomes are able to activate bone marrow stromal cells which in turn release IL 8 acting as an in vitro and in vivo pro survival factor for chronic myelogenous leukemia cells.	Corrado et al. (2014)	
The NKG2D ligands (MICA/B, ULBP1, ULBP2), and HSP70	The human T cell leukemia Jurkat- and B cell leukemia/lymphoma Raji cell lines	qRT-PCR analysis, Flow cytometry assay, and Western blotting	The NKG2DL-carrying exosomes abrogate NKG2D-mediated NK-cell cytotoxicity and, thus, might contribute to the immune evasion of leukemia/lymphoma cells T- and B-cell lines Jurkat and Raji as hematopoietic malignancy models.	Hedlund et al. (2011)	
CD40, CD86, HSP60, HSP70, HSP90, RANTES, and IL-1b	The six-to-eight week old female BALB/c (H-2d) and C57BL/6J (H-2b) mice
The mouse B cell lymphoma/leukemia cell line A20 (H-2d) cells
The colon tumor 26 (CT-26) of colon adenocarcinoma in BALB/c mice	Antigen presentation assay, Flow cytometry analysis, Enzyme-linked immunosorbent assay (ELISA), and Western blotting	The exosomes derived from heat-shocked lymphoma cells contain more HSP60 and HSP90 and increased amounts of molecules involved in immunogenicity including MHC class I, MHC class II,
CD40, CD86, RANTES and IL-1b. Consistent with the in vitro results the HS-Exo exhibit a more potent antitumor effect than control exosomes in prophylaxis and therapeutic in vivo lymphoma models.
	Chen et al. (2006)	
HLA class I and II molecules such as HLA-B, HLA-C histocompatibility antigen, B-15 alpha chain, B-39 alpha chain, A-26 alpha Chain, HLA-DQA1 MHC
class II antigen, HLA class II histocompatibility antigen,
DQ(1) beta Chain, HLA-C antigen, Cw-4 alpha and Cw-3
alpha chain, HLA-DPB1 major histocompatibility complex,
class II, DP beta1, CD19, CD20, CD22, CD81, CD82, antigen and intercellular adhesion molecule 1, etc	The human B cell leukemia/lymphoma Raji cell lines	Mass spectrometry assay, and Western blotting	The lymphoma cell-derived exosomes (LCEXs) expressed a discrete set of proteins involved in antigen presentation and cell migration and adhesion, indicating that LCEXs play a significant role in the regulation of immunity and interaction between lymphoma cells and their microenvironment.	Yao et al. (2015)	
ALIX, TSG-101, CD63, CD9, CD81, CD24, HSP70, and HSP90	The syngeneic BALB/c T-cell lymphoma cell line LBC (H-2d) cells
Six- to ten-week-old female immunocompetent BALB/c mice	Flow cytometry analysis, Enzyme-linked immunosorbent assay (ELISA), Dot blot and Western blotting	T-cells from EVs A-immunized mice secreted IFN-γ in response to tumor stimulation. Thus, tumor-specific CD4+ and CD8+ IFN-γ secreting cells could be effectively expanded from mice immunized with EVs A, indicating that a T helper 1 response is associated with tumor rejection.	Menay et al. (2017)	
CD63, CD81, CD19, CD20, CD22, CD23, CD24, CD37, CD40, and CD45	The B-cell lymphoma cell lines Ramos, SUDHL-4, SUDHL-6, and Ros-50 cells
The colon adenocarcinoma SW480 cell line	qRT-PCR analysis, Flow cytometry assay, Electron microscopy assay, and Western blotting	The several B-cell surface antigens including CD19, CD20, CD24, CD37, and HLA-DR, but not CD22, CD23, CD40, and CD45 are expressed on exosomes from B-cell lymphoma cell lines with large heterogeneity among the different B-cell lymphoma cell lines. Interestingly, these B-cell lymphoma–derived EVs are able to rescue lymphoma cells from rituximab-induced complement-dependent cytotoxicity.	Oksvold et al. (2014)	
CD63, CD81, CD20, CD19, MCL4, MCL8, and MCL7	The Mantle cell lymphoma cell lines Jeko-1, and Mino cells
The Jurkat human acute T cell leukemia cell line and HS-5 human bone marrow derived stroma cell line
The blood plasma of MCL patients and healthy donors	qRT-PCR analysis, Flow cytometry assay, Electron microscopy assay, Nanoparticle tracking analysis (NTA), and Western blotting	The MCL exosomes are quickly and preferentially internalized via B-lymphocytes. Only minor fraction of exosomes was internalized into T-cell leukemia and bone marrow stroma cell lines, when these cells were co-cultured with MCL cells. Thus, exosome internalization was not suppressed by specific siRNA against caveolin1 and clathrin but was found to be mediated by cholesterol-dependent pathway.	Hazan-Halevy et al. (2015)	
MiR-9, MiR-146a, and MiR-155	The human Burkitt’s lymphoma cell lines Raji, and Ramos cells
The retinal pigment epithelial cell line ARPE-19 cells
The human umbilical vein endothelial cells (HUVECs)	qRT-PCR analysis, Enzyme-linked immunosorbent assay (ELISA), and Western blotting	Raji-exosome mediated delivery of miR-155 inhibitor diminished endogenous and secreted levels of VEGF-A in ARPE-19 cells. Also, a significant increase in cellular levels of miR-9, miR-146a, and miR-155 in co-cultures of Raji cell compared with EBV-negative B cells was detected.	Yoon et al. (2016)	
Wnt3a, and SFRP4	The diffuse large B-cell lymphoma (DLBCL) cell lines SUDHL4, U2932, OCI Ly1, OCI Ly3, and Karpas 422 cells
The human B-lymphocytic lymphoma cell lines BALM - 3 cells
The spontaneously immortalized cell line L-Wnt3a cells	qRT-PCR analysis, Flow cytometry assay, and Western blotting	The diffuse large B-cell lymphomas possessed a self-organized infrastructure comprising side population (SP) and non-SP cells, where transitions between clonogenic states are modulated by exosome mediated Wnt signaling. Lymphoma SP cells displayed autonomous clonogenicity and exported Wnt3a via exosomes to neighboring cells, thus modulating population equilibrium in the tumor.	Koch et al. (2014)	
MiR-96-5p, MiR-182-5p, and MiR-149	The human colon carcinoma cell lines HT-29 and HCT-116 cells
The peripheral fasting blood specimens of colon carcinoma patients
The human colon carcinoma tissues and normal tissue samples	qRT-PCR analysis, Flow cytometry assay, and Western blotting	The considerably promoted GPC1+ exosomes are present in the plasma of CRC patients and can be released from CRC tumor cells. The high expression of miR-96-5p and miR-149 significantly decreased cell viability and enhanced cell apoptosis in HT-29 and HCT-116 cells, and suppressed the growth of xenograft HT-29 and HCT-116 tumors.	Li et al. (2017)	
Dickkopf-related protein 4 (DKK4)	The human colon carcinoma cell lines SW480 and SW480APC	qRT-PCR analysis, Electron microscopy assay, and Western blotting	The secretion of Wnt antagonist, dickkopf-related protein 4 (DKK4) enhanced in SW480APC colon carcinoma cells derived exosomes. In addition, the promoter region of the DKK4 gene appears to have decreased methylation in SW480APC cells, comparing with the paternal SW480 cells, as well as reduced expression of DNA methyltransferase 3a (DNMT-3a).	Lim et al. (2012)	
Tumor suppressor-activated pathway 6 (TSAP6)	The human colon carcinoma cell line HCT-116 TP53-wild type, and HCT-116
TP53-null cells
The human colon carcinoma tissues and normal tissue samples
The peripheral fasting blood specimens of colon carcinoma patients	qRT-PCR analysis, Flow cytometry assay, and Western blotting	The expression of TSAP6 is not related with release of exosomes; and regulation of TSAP6 through P53 was not detected either in tumor samples or in HCT-116cell lines. Besides, it was not shown that the P53/TSAP6 pathway regulates the release of exosomes into the plasma of colorectal cancer patients.	Silva et al. (2012)	
CD9, CD97, ERK, JNK, p38, HSP70, MiR-2861, MiR-4734, MiR-4728-5p, MiR-6165	The stomach adenocarcinoma cell line SGC-7901 cells	qRT-PCR analysis, Electron microscopy assay, and Western blotting	CD97 elevates gastric cancer cell proliferation and invasion in vitro via exosome-mediated MAPK signaling pathway, and also exosomal miRNAs including miR-2861 and miR-4734 are probably involved in activation of the CD97-associated pathway.	Li et al. (2015)	
fibronectin 1 (FN1), and
laminin, gamma 1 (LAMC1)	The human gastric cancer cell lines KatoIII, MKN45, and MKN74 cells
The human normal mesothelial
cell line MeT-5A cells	qRT-PCR analysis, and Western blotting	The expression of adhesion-related molecules, including fibronectin 1 (FN1) and laminin gamma 1 (LAMC1), were promoted in mesothelial cells after internalization of tumor-derived exosomes (TEX) from gastric cancer cell line and malignant pleural effusion.	Arita et al. (2016)	
Epidermal growth factor receptor (EGFR)	The human GC liver metastatic and paired adjacent non-cancerous tissues
Male nude mice (BALB/c-nu, 6 to 8 weeks)
The human gastric adenocarcinoma cell line SGC7901 cells
The primary mouse liver cells were obtained from the livers of C57BL/6J mice (6–8 weeks of age)	qRT-PCR analysis, Enzyme-linked immunosorbent assay (ELISA), Electron microscopy assay, Nanoparticle tracking analysis (NTA), and Western blotting	The EGFR-containing exosomes derived from cancer cells is demonstrated to impressively activate hepatocyte growth factor (HGF) by inhibiting miR-26a/b expression. In addition, the high expressed of paracrine HGF, which binds the c-MET receptor on the migrated cancer cells, provides fertile ‘soil’ for the ‘seed’, simplifying the landing and proliferation of metastatic cancer cells.	Zhang et al. (2017)	
HLA-A, and CD9	The human gastric adenocarcinoma cell line SGC7901 and Jurkat T cells	qRT-PCR analysis, Electron microscopy assay, Western blotting and Immunoprecipitation	The Cbl family of ubiquitin ligases might be involved in regulation of exosome-induced apoptosis of Jurkat T cells by promoting PI3K proteasome degradation, inactivation of PI3K/Akt signaling, and mediating some effects of caspase activation.	Qu et al. (2009)	
MiR-203, MiR-212-3p
Several proteins (attractin, complement proteins C3, C4 and C5, integrin, and lactotransferrin)	The human pancreatic carcinoma epithelial like cell line PANC-1 cells	Liquid chromatography-electro spray ionization mass spectrometry/mass spectrometry (LC-ESIMS/MS) analysis, Enzyme-linked immunosorbent assay (ELISA), and Western blotting	The pancreatic cancer (PC)-derived exosomes down regulated the expression of TRL4 in dendritic cells (DCs) through miR-203, including immune tolerance. Therefore, the PC-derived exosomal miRNAs can down regulate the anti-tumor activity of DC/cytokine-induced killer cells (CIKs) and that depletion of exosomal miRNAs can promote the anti-tumor activity of DC/CIKs.	Que et al. (2016)	
CD63, TSG101, and Alix	The pancreatic cancer patient-derived cell lines 6741-1 (MCPAN014), 6413-1 (MCPAN013), 5822-1 (MCPAN008), 7135-1, 7426-1, and 7291-1 cells
Murine 3T3-L1 preadipocytes
The blood collection and tissue samples of pancreatic cancer patients	Enzyme-linked immunosorbent assay (ELISA), Immunoprecipitation, and Western blotting	Lipolysis in 3T3-L1 cells and in human adipocytes enhanced upon exposure to PC-exosomes. Increase in lipolysis is attributed to adrenomedullin (AM) contained within PC-exosomes, as AM receptor blockade led to abrogation of the effect of exosomes and activation of ERK1/2 and p38 MAPKs in both murine and human adipocytes.	Sagar et al. (2015)	
HSP70	The MIA Paca-2, an epithelial-like pancreatic cancer cell line	Electron microscopy assay, Western blotting	The EXO/SEB is a novel model or apopto-immunotherapy, being able to induce apoptosis in addition to specific immune responses. The enhanced expression of antiapoptotic genes including Bax, Bak and fas in cells treated with the EXO/SEB causes promotion of apoptosis. In addition, EXOs released from pancreatic cancer cells can trigger the mitochondrial-dependent apoptosis and increase the caspase-3 and caspase-9 activities.	Mahmoodzadeh et al. (2014)	
Alix, TSG101, CHMP4B, and ATP-binding cassette sub-family G member 2 (ABCG2)	The human pancreatic cancer cell lines AsPC-1 and PANC-1	qRT-PCR analysis, Electron microscopy assay, Western blotting, and Proteomics analysis	The involvement of GIPC on metabolic stress pathways regulating autophagy and microvesicular shedding, and observed that GIPC status determines the loading of cellular cargo in the exosome. Thus, the detection showed the overexpression of the drug resistance gene ABCG2 in exosomes from GIPC-depleted pancreatic cancer cells.	Bhattacharya et al. (2014)	
KRAS	Blood plasma samples of 39 early-stage pancreatic ductal adenocarcinoma (PDAC) patients and 82 agematched healthy controls	qRT-PCR analysis, Flow cytometry assay, Electron microscopy assay, and Western blotting	By comparing exoDNA to cfDNA in liquid biopsies of patients with pancreatic ductal adenocarcinoma, the higher exoKRAS mutant allele frequency, but not CA19-9, was associated with disease free survival in patients with localized disease.	Allenson et al. (2017)	
CD44v6, Tspan8, EpCAM, MET, CD104, CD184, Tspan8, CD24, CD133, CD9, CD63, CD151, MiR-1246, MiR-4644, MiR-3976, and MiR-4306	Blood collection samples from 131 PaCa, 25 chronic pancreatitis (CP), 22 benign pancreatic tumor and 12 patients with non-PaCa, and 30 volunteers
The human PaCa tumor cell lines AsPC1, Capan1, Panc1, ExPC3, A818 cells	qRT-PCR analysis, Flow cytometry assay, and Microarray miRNA analysis	MiR-1246, miR-4644, miR-3976, and miR-4306 were significantly upregulated in 83% of PaCa serum-exosomes, but rarely in control groups. These miRNA were also elevated in exosome-depleted serum of patients with PaCa, but at a low level. Also, the expression of the PaCIC markers CD24, CD44v6, CD104, Tspan8, EpCAM, MET, and CD151 and the common exosome markers CD9 and CD63 was based on high expression in tumor tissue.	Madhavan et al. (2015)	
Alix, CD9, CD63, CD81, syntenin, calreticulin, calpain 1, VDAC1, vimentin, hepatoma-derived growth factor, casein kinase II α, and annexin A2	The human urinary bladder transitional cell carcinoma cell lines T24, FL3,
and SLT4 cells	LC-MS/MS analysis, Electron microscopy assay, and Western blotting	The several proteins lead to EMT was detected in bladder carcinoma cells, including enhanced abundance of vimentin and hepatoma-derived growth factor in the membrane, and casein kinase IIα and annexin A2 in the lumen of exosomes, respectively, from metastatic cells. The change in exosome protein abundance correlated little, although significant for FL3 versus T24, with alters in cellular mRNA expression.	Jeppesen et al. (2014)	
lncARSR (lncRNA Activated in RCC with Sunitinib Resistance), HSC70, ALIX, CD43, heterogeneous nuclear ribonucleoprotein
A2B1 (hnRNPA2B1), TSPAN8, VPS36, and CD63	The nude mice grafted with 786-O and
ACHN cells
The renal cell carcinoma (RCC) cell lines 786-O, and ACHN
The sunitinib-resistant RCC cells 7Su3rd, 771S, 771R, ACSu3rd	qRT-PCR analysis, Electron microscopy assay, and Western blotting	In sunitinib-resistant renal cell carcinoma (RCC) cells, IncRNA activated in RCC with sunitinib resistance (lncARSR) elevates sunitinib resistance by competitively binding miR-34 and miR-449, leading to the increased expression of AXL/c-MET and reactivation of STAT3, AKT, and ERK signaling. Moreover, lncARSR can be packaged into exosomes and secreted from sunitinib-resistant RCC cells, transferring resistance to recipient-sensitive cells.	Qu et al. (2016a)	
MiR-34a, MiR-141, MiR-134, MiR-135a, MiR-135b, and MiR-370	The adenocarcinomic human alveolar basal epithelial cell line A549 cells
The collection of non-small cell lung cancer (NSCLC) patient tissue samples	qRT-PCR analysis, Electron microscopy assay, and Western blotting	YKT6 downregulation is associated with a remarkable reduction in exosome release in an NSCLC cell line and that low YKT6 expression is associated with better clinical outcome in NSCLC patients. Thus, YKT6 is a SNARE protein in the regulation of exosome release in lung cancer cells and is in turn accurately regulated by miR-134 and miR-135b.	Ruiz-Martinez et al. (2016)	
MiR-378a, MiR-379, MiR-139- 5p, MiR-200b-5p, MiR-151a-5p, MiR-30a-3p, MiR-200b-5p, MiR-629, MiR-100, and MiR-154-3p	30 blood plasma samples (10 patients affected by lung adenocarcinomas, 10 with lung granulomas, and 10 healthy smokers)
	qRT-PCR analysis	The production of exosomes containing miRNAs in the lung carcinoma cells are completely different to those present in healthy control cells from which neoplastic cells originated.	Cazzoli et al. (2013)	
MiR-17-3p, MiR-21, MiR-106a, MiR-146, MiR-155, MiR-191, MiR-192, MiR-203, MiR-205, MiR-210, MiR-212, and MiR-214	Plasma samples from patients with lung adenocarcinoma and a control group without known lung cancer or other active cancer	Microarray analysis	The considerable difference in total exosome and miRNA levels between lung cancer patients and controls, and the similarity between the circulating exosomal miRNA and the tumor-derived miRNA patterns, suggest that circulating exosomal miRNA might be useful as a screening test for lung adenocarcinoma.	Rabinowits et al. (2009)	
Curcumin (anti-cancer drug for lung cancer remedy)	The human lung cancer cell lines BEAS-2B, A549, PC9, and H1299 cells	qRT-PCR analysis, and Western blotting	The anti-cancer effects of Curcumin are associated with upregulation of transcription factor 21 (TCF21), mediated by downregulation of DNMT1. Also, TCF21 overexpression and knockdown was introduced to H1299 cells through lentiviral system, which led to suppression and promotion of lung tumor growth, respectively.	Wu et al. (2016)	
CD63, flotillin-1, and HSP70	The human lung adenocarcinoma cell lines A549, and H460 cells
BALB/c nude male mice	qRT-PCR analysis, Flow cytometry assay, Electron microscopy assay, and Western blotting	The β-elemene significantly suppressed growth and induced apoptosis in lung cancer cells. The levels of the anti-apoptotic genes Bcl-2 and Bcl-xl in A549 cells decreased, while expression of P53 and production of exosomes, and the exosome markers CD63, flotillin-1, and HSP70 increased after β-elemene remedy.	Li, Liu & Wang (2014)	
CD63, Calnexin, MiR-122, MiR-126, MiR-128, MiR-143, MiR-144, MiR-302a, MiR-302c	The consecutive series of blood and bronchoalveolar lavage (BAL) samples from 30 non-small cell lung cancer (NSCLC) patients and 75 patients with non-tumor pathology	qRT-PCR analysis, MicroRNA Quantitative PCR Array, and Western blotting	Exosome levels were considerably higher in plasma than in bronchoalveolar lavage (BAL) samples in both groups of patients. Also, in tumor patients the number of miRNAs with high expression was greater in the exosomes released to plasma than in those released to the airway.	Rodríguez et al. (2014)	
Epidermal Growth Factor Receptor (EGFR)	The human lung carcinoma cell lines HARA, HARA-B, A549, RERF-LC-MS and LU65 cells
Primary human pulmonary alveolar epithelial cell line HPAEpiC cells
Blood plasma samples from tumor-bearing mice, and the human blood plasma specimens from both healthy individuals and cancer patients
Six-week-old male BALB/c Slc-nu/nu mice	Enzyme-linked immunosorbent assay (ELISA), Electron microscopy assay, and Western blotting	The secretion of exosomes in plasma that express high levels of EGFR are clearly derived from tumor tissue samples. Also, the exosomal EGFR detection could potentially be applied in blood tests to diagnose lung cancer because the exosomal EGFR level was higher in lung cancer patients than in normal Individuals.	Yamashita et al. (2013)	
CD9, and CD63	The human lung carcinoma cell lines H1299 and H522 cells
The human pulmonary alveolar epithelial cells	Principal component analysis (PCA), Enzyme-linked immunosorbent assay (ELISA), Electron microscopy assay, and Western blotting	The experiment showed the successful segregation of NSCLC-derived exosomes from normal alveolar cell-derived exosomes using the noninvasive method of SERS accompanied with PCA. Therefore, the Raman signals of lung cancer cell derived exosomes and normal alveolar cell-derived exosomes are well distinguished through PCA.	Park et al. (2017)	
Polyadenylate-binding protein 1 (PABP1)	The human duodenal cancer cell line HuTu 80 cells
The human gastric cancer cell line AZ-521 cells, and the metastatic gastric cancer cell line AZ-P7a cells	qRT-PCR analysis, and Western blotting	The PABP1 is predominantly abundant in exosomes from a metastatic duodenal cancer cell line even though its intracellular expression levels do not vary among cell lines. Thus, AZ-P7a cells do not tolerate intracellular PABP1 accumulation and are thus supported into the extracellular milieu through the exosome –mediated pathway.	Ohshima et al. (2014)	
MiR-21	The blood plasma samples of esophageal squamous cell carcinoma (ESCC) and healthy volunteers
The human esophageal cancer cell line EC9706 cells	qRT-PCR analysis, Flow cytometry assay, Microarray analysis, and Western blotting	The Cy3-labeled miR-21 mimics could be transferred between esophageal cancer cells by exosomes. Thus, the miR-21 mimics could affect migration and invasion of recipient cells partly via modulation of its target gene PDCD4 and its downstream-signaling molecules, MMP-2 and MMP-9 by using the cell co-culture system. Also, miR-21 was upregulated significantly in plasma from esophageal cancer patients and indicated a significant risk association for esophageal cancer.	Liao et al. (2016)	
HSP27	The human ovarian cancer (OC) cell lines OVCAR-3 and SK-OV-3 cells	Enzyme-linked immunosorbent assay (ELISA), and Western blotting	The heat shock protein HSP27 has been correlated in OVCAR-3 and SK-OV-3 cells ovarian cancer cell lines by exosomes with aggressiveness and chemoresistance and, thus, represents a promising potential biomarker for OC diagnosis, prognosis, and treatment response.	Stope et al. (2017)	
CA-125, EpCAM, and CD24	The blood plasma samples of ovarian cancer (OC) patients	Flow cytometry assay, Enzyme-linked immunosorbent assay (ELISA), Electron microscopy assay, and Western blotting	Through the exosome analysis enabled by the ExoSearch chip has been applied for ovarian cancer diagnosis via quantifying a panel of tumor markers from exosomes in a small-volume of blood plasma (20 μL), which indicated significant diagnostic accuracy and was comparable with standard Bradford assay.	Zhao et al. (2016)	
MiR-584, MiR-517c, MiR-378, MiR-520f, MiR-142-5p, MiR-451, MiR-518d, MiR-215, MiR-376a, MiR-133b, and MiR-367	The human Hepatocellular carcinoma (HCC) cell lines Hep3B, HepG2, and PLC/PRF/5 cells	qRT-PCR analysis, , Flow cytometry assay, and Electron microscopy assay	The HCC cell-derived exosomes can modulate β activated kinase-1 (TAK1) expression and associated signaling and promote transformed cell growth in recipient cells. Loss of TAK1 has been implicated in hepatocarcinogenesis and is a biologically plausible target for intercellular modulation.	Kogure et al. (2011)	
HSP60, HSP70, and HSP90	The human hepatocellular
carcinoma cell line HepG2 and PLC/PRF/5 cells
The erythromyeloblastoid leukemia cell line K562 cells
The blood plasma samples of hepatocellular carcinoma cells (HCC) patients and healthy donors	Enzyme-linked immunosorbent assay (ELISA), Flow cytometry assay, Electron microscopy assay, and Western blotting	The anti-cancer drugs (Paclitaxel, Etoposide, Carboplatin, Irinotecan hydrochloride, Mitoxantrone hydrochloride, Epirubicin hydrochloride, Cisplatin, Mitomycin, Fluorouracil, Oxaliplatin, and Gemcitabine hydrochloride) can efficiently up-regulate the expression of HSPs (HSP60, HSP70, and HSP90) on the human hepatocellular carcinoma cell-derived exosomes and the ability of exosomal HSPs as a tumor vaccine to significantly induce NK cells reacts that lead to eliciting an anti-tumor immune response in vivo.	Lv et al. (2012)	
CD10, CD26, CD81, PrPc, and Slc3A1	The urine samples obtained from experimental models of mouse, and male Wistar rats, 14 week of age	NanoLC-MS/MS analysis, Electron microscopy assay, and Western blotting	The enhancement in the level of CD10 protein was detected in urinary exosomes obtained from glycine N-methyltransferase knockout mice, an animal model of chronic liver injury associated with steatosis, fibrosis, and human Hepatocellular carcinoma (HCC). In addition, the proteome of different vesicle populations indicates several biomarkers including PrPc, Cd26, Slc3a1, Cd81, and Cd10 that are detected in urinary vesicles and may be useful for diagnostic purposes.	Conde-Vancells et al. (2010)	
MiR-16-1, MiR-21, MiR-24, MiR-31, MiR-122, MiR-125b, MiR-223, MiR-410, CD29, and CD44	The human adult liver stem cells (HLSC) were isolated from human cryopreserved normal hepatocytes
The normal human hepatocytes and human fibroblasts
The hepatoma cell line HepG2 cells	qRT-PCR analysis, and Western blotting	The microvesicles (MVs) derived from HLSC suppressed the growth of hepatoma tumors and cell line HepG2 cells by transferring the genetic information and delivering anti-tumor miRNAs that interfered with the deregulated survival and proliferation of these cells.	Fonsato et al. (2012)	
EIF2C2 (AGO2), CHEK2, CDK2, and MATK	The human adult liver stem cells (HLSC) were isolated from human cryopreserved normal hepatocytes
Rat liver tissue samples	qRT-PCR analysis, Enzyme-linked immunosorbent assay (ELISA), and Western blotting	The microvesicles (MVs) derived from HLSC may activate a proliferative program in remnant hepatocytes after hepatectomy through a horizontal transfer of specific mRNA subsets. The MVs-mediated transfer of mRNA from HLSC to hepatocytes can display a procedure that contribute to liver regulation and that could be extracted in regenerative medicine.	Herrera et al. (2010)	
MiR-92a, and MiR-638	The hepatocellular carcinoma (HCC) cell lines HepG2, OR6 and SN1a cells
The blood plasma samples of hepatocellular carcinoma cells (HCC) patients and healthy donors	qRT-PCR analysis, and MTT assay	The miR-92a is highly expressed in hepatocellular carcinoma (HCC). Thus, the expression level of miR-92a affects the proliferation of hepatoma cell lines HepG2, OR6 and SN1a cells. Also, the ratio of miR-92a/miR-638 decreased in the plasma samples from the HCC patients compared with healthy donors.	Shigoka et al. (2010)	
CD9, and CD63	The human HCC cell lines SMMC-7721, MHCC-97H, MHCC-97 L, and LO2 cells
4 to 6 week old male BALB/c nu/nu mice	MTT assay, Enzyme-linked immunosorbent assay (ELISA), Electron microscopy assay, Fluorescence-activated cell sorting (FACS) analysis, and Western blotting	The HCC cell-derived exosomes mediate Sorafenib resistance in HCC cells in vitro, and exosomes derived from highly invasive tumors have grater effects than those derived from less invasive tumors. Also, HCC cell-derived exosomes exerted their functions through enhancing the level of proteins associated with Sorafenib resistance, protecting tumor cells from Sorafenib-induced apoptosis and activating the HGF/c-Met/Akt signaling pathways in vitro.	Qu et al. (2016b)	
CD63, tumor susceptibility gene-101 (TSG-101)	Male Fischer-344 (F344) rats
N1S1 rat HCC cells (hepatocellular carcinoma cells)
Blood plasma samples from the Lateral tail vein of the Rat	Electron microscopy assay, and Western blotting	The adipose-derived mesenchymal stem cells (ADMSCs) derived exosomes enhanced natural killer T-cell (NKT) cell anti-tumor response in rats, through facilitating HCC inhibition, early apparent diffusion coefficient (ADC) increase, and low-grade tumor differentiation.	Ko et al. (2015)	
ACTB, TUBA1A, FN1, FNLA, CD61, HLA-A, LGALS3BP, Alix, RAB5B, RAB5C, SDCBP, VPF37B, CLTC, ARF1, ANXA2, ANXA5, HSC70, HSP72, RAC1, STOM, MFGE8, MVP, GNA12, PTGFRN, HBA1, tumor susceptibility gene-101 (TSG-101), and Grp94	The inhoused established human HCC cell lines HKCI-C3 and HKCI-8 cells
The hepatocellular carcinoma (HCC) cell line MHCC97L cells
The immortalized hepatocyte cell line MIHA cells	qRT-PCR analysis, Ion Torrent Next-Generation Sequencing, and Western blotting	The internalization of exosomes could activate PI3K/AKT and MAPK signaling pathway, and promote secretion of MMP-2 and MMP-9 that favored cell invasion. Also, by proteome analysis Syndecan–syntenin–ALIX is known to support biogenesis of exosomes and the segregation of signaling cargo to these vesicles. The research also detected the components of endosomal protein sorting complex, such as VPS28 and VPS37, whose functions are required for exosome cargo sorting Process.	He et al. (2015)	
MiR-718, and MiR-1246	Six cases that underwent living donor liver transplantation (LDLT)
Blood plasma samples from patients that underwent living donor liver transplantation (LDLT)
The hepatocellular carcinoma (HCC) cell lines Huh-7, and PLC/PRF/5 cells	qRT-PCR analysis, Electron microscopy assay, MTT assay, MicroRNA microarray analysis, and Western blotting	The specific biomarker miR-718 showed significantly different expression in the serum exosomes of HCC cases with recurrence after LT compared with those without recurrence. Decreased expression of miR-718 was associated with HCC tumor aggressiveness in the validated cohort series.	Sugimachi et al. (2015)	
MiR-21, CD63, and tumor susceptibility gene-101 (TSG-101)	Blood plasma samples from the hepatocellular carcinoma (HCC), and hepatitis B (CHB) patients	qRT-PCR analysis, Electron microscopy assay, and Western blotting	The expression of serum exosomal miR-21 was higher in patients with HCC than in patients with CHB and healthy volunteers, the sensitivity of detection is much lower than using exosomal miR-21. These findings indicate that miR-21 is enriched in serum exosomes which provides increased sensitivity of detection than whole serum.	Wang et al. (2014a)	
HSP70, major histocompatibility complex (MHC) class I, polypeptide-related sequence A (MICA) and MICB	Peripheral blood samples of hepatocellular carcinoma (HCC) patients
Peripheral blood mononuclear cells (PBMCs)
Peripheral blood NK cells
The hepatocellular carcinoma (HCC) cell lines HepG2 cells	qRT-PCR analysis, Electron microscopy assay, and Western blotting	MS-275 (one of the histone deacetylase inhibitor (HDACi) drugs) modified exosomes enhance the cytotoxic effect of NK cells significantly through upregulating the expression of MICA, MICB and HSP70.	Xiao et al. (2013)	
MiR-10b, MiR-21, MiR-122, and MiR-200a	The hepatocellular carcinoma (HCC) tissues and blood plasma samples of 108 male fisher 344 rats	qRT-PCR analysis, Electron microscopy assay, Flow cytometry assay, and Western blotting	The changing in the expression of both exosomes and miRNAs (miR-10b, miR-21, miR-122, and miR-200a) was observed during cirrhosis, which in contrast with alpha-fetoprotein (AFP) starts showing up until the early HCC stage. Therefore, the combination of circulating miRNAs and exosomes might serve as promising biomarkers for non-virus infected HCC screening and cirrhosis discrimination.	Liu et al. (2015)	
Transactive response DNA-binding protein of 43 kDa (TDP-43)	The human glioma cell line U251 cells
Collection of cerebrospinal fluid (CSF) from glioblastoma patients and normal control (NC)-CSF	Western blot analysis	The ALS-FTD-CSF incubation with U251 cells generate TDP-43 mislocalization, prion-like propagation of TDP-43 aggregates, and the cell–cell transmission of TDP-43 accumulates is mediated through exosome and TNTs-like structure. Thus, incubation of ALS-CSF and ALS-FTD-CSF with U51 causes toxic to the cells.	Ding et al. (2015)	
TrkB, P75NTR, sortilin, HSP90, CD63, and CD9	The human GBM cell line U87-MG cells
Two derived cell lines from U87-MG cells (human empty vector pLKO, control cells and sh YKL-40 cells)
Female NOD/SCID mice
Blood plasma samples of glioblastoma patients	qRT-PCR analysis, Electron microscopy assay, Flow cytometry assay, and Western blotting	The loss of aggressiveness in YKL-40-silenced cells significantly reduced TrkB, p75NTR and sortilin expression. Thus, the release of TrkB in exosomes from control glioma cells, was able to rescue both migration and activation of YKL-40-inactivated cells.	Pinet et al. (2016)	
_	Luciferase expression mice glioblastoma cell line GL26- Luc cells
C57BL/6j mice (H-2b) (6–8 weeks, female)	Flow cytometry assay	The GL26 cells-derived exosomes significantly promote GL26 tumor growth in vivo. In fact the GL26 cells-derived exosomes, suppressed the cytotoxic activity of CD8+T cells both in vivo and in vitro, that leading to reduction of CD8+T cells in spleen and inhibition of cytolytic associated IFN-γ and granzyme B.	Liu, Wang & Yuan (2013)	
MiR-9, CD44, CD45, CD105	The glioblastoma multiforme (GBM) tissue samples
The glioblastoma cell lines U87, T98G, BT145 (primary GBM), and BT164 (recurrent GBM) cells
The mesenchymal stem cells (MSCs)	qRT-PCR analysis, Flow cytometry assay, and Western blotting	The promotion of miR-9 elevates temozolomide (TMZ)-resistant GBM cells. To block miR-9, methods were developed with Cy5-tagged anti-miR-9. Dye-transfer studies indicated intracellular communication between GBM cells and MSCs. This occurred by gap junctional intercellular communication and the release of microvesicles. Thus, anti-miR-9 was transferred from MSCs to GBM cells.	Munoz et al. (2013)	
CRYAB (crystallin, alpha B), CD9, CD63	The human glioma cell line U373 cells	Enzyme-linked immunosorbent assay (ELISA), Electron microscopy assay, and Western blotting	Increase in CRYAB levels in GBM coupled with its secretion via exosomes points to an important mode of intercellular communication which, in GBM, may confer resistance to apoptosis in surrounding cells following radiation and chemo-therapies. Proinflammatory cytokines also bring about profound changes in the proteome of the exosome.	Kore & Abraham (2014)	
CD11b, CD14, CD16, and CD163	The glioblastoma multiforme (GBM) tissue samples
The human peripheral blood mononuclear cells (PBMC) of GBM patients	Enzyme-linked immunosorbent assay (ELISA), Flow cytometry assay, and Luminex Analysis	The M2-like monocytes expressing CD14+ and CD163+, another indicator of Th2 bias, are promoted in GBM patient blood and associated with high serum concentrations of colony 2 stimulating factor 2 and 3, as well as interleukin-2, -4, and -13, the latter 2 cytokines being hallmarks of Th2 immunity. Fractionation of GBM patient sera into samples enriched for exosomes or soluble factors proved that both fractions are capable of inducing CD163 expression in normal monocytes.	Harshyne et al. (2015b)	
Nanofilament	The human glioblastoma cell lines U87 and U251cells
The human melanoma cell line SKMEL cells
Normal human astrocytes (NHA)	Piezoresponse force microscopy (PFM)	Compared with normal exosomes, glioblastoma exosomes displayed numerous nanofilaments, and the nanofilaments were trypsin- and RNase-resistant. Based on in vitro uptake assays, glioblastoma exosomes indicated a significantly higher uptake in cells compared with normal exosomes.	Sharma et al. (2014)	
Actin, CD9, CTGF, tumor susceptibility gene 101 (TSG-101), apoptosis-linked gene2-interacting protein x (Alix), IGFBP2, phospho-/total TrkA, phospho-/total FAK, phospho-/total src, phospho-/total Paxillin	The glioblastoma multiforme (GBM) cell lines LN18, U87MG, and U251cells
The GBM stem-like cell lines GBAM1, and GBMJ1 cells
The Human umbilical vein endothelial cells (HUVEC-CS)	qRT-PCR analysis, Flow cytometry assay, Electron microscopy assay, and Western blotting	CTGF mRNA and IGFBP2 protein levels were elevated, and coculture of nonirradiated cells with exosomes isolated from irradiated cells increased CTGF protein expression in the recipient cells. Besides, these exosomes promoted the activation of TrkA, FAK, Paxillin, and Src in recipient cells, molecules involved in cell migration.	Arscott et al. (2013)	
CD63, CD71, CD81, and AS-ODN (Antisense oligodeoxynucleotide)	The human glioblastoma cell line U118 cells
Primary patient tumor cell lines
The human peripheral
blood mononuclear cells (PBMC)
Murine glioma cell line GL261	Flow cytometry assay, Electron microscopy assay, and Enzyme-linked immunosorbent assay (ELISA)	The included AS-ODN releasing from the chambers is directed against the insulin-like growth factor type-1 receptor, is immunostimulatory, and therefore leading to promote presentation of these antigens. The glioma-derived exosomes were detected to express CD63, CD71, and CD81, endosomal antigens.	Harshyne et al. (2015a)	
CD 9, CD63, and CD81	The brain neuronal glioblastoma-astrocytoma cell line U-87 MG cells
The Immortalized mouse brain endothelial cell line bEND.3 cells
The neuroectodermal tumor cell line PFSK-1 cells
The glioblastoma cell line A-172 cells	qRT-PCR analysis, Flow cytometry assay, Electron microscopy assay, Enzyme-linked immunosorbent assay (ELISA), and Western blotting	The exosomes released from brain endothelial cells delivered anticancer drug across the blood-brain barrier (BBB), which subsequently exerted cytotoxic efficacy against brain cancer. Also, the high presence of CD63 in bEND.3 exosomes indicates that these exosome nanovesicles might be differently implicated in receptor-mediated transport across the BBB.	Yang et al. (2015)	
Collagen type VI alpha 1, putative RNA-binding protein 15B chain A, substrate induced remodeling of the active site regulates HTRA1, coatomer protein complex-subunit beta 2, myosin-heavy chain 1, keratin-type I cytoskeletal 9, HSP90, and CD63	The brain neuronal glioblastoma-astrocytoma cell line U-87 MG cells	Electron microscopy assay, matrix-assisted laser desorption ionization time-of-flight mass spectrometry (MALDI-TOF-MS) analysis, and Western blotting	Through the proteome analysis of U-87MG exosome the Hsp90 was promoted in exosomes exposed to a low temperature compared with exosomes incubated under normal conditions. Also, there was detected an increase expression of calcium-dependent secretion activator 2 isoform b, hCG1817425, armadillo repeat-containing protein 4, and immunoglobulin heavy variable 5-a in low temperature-exposed proteome. Besides, the proteins that were reduced on the L.T. gel were collagen alpha-1(VI), DNA topoisomerase I, titin, mitochondrial isoform 2, RNA-binding protein 15B, phosphoserine aminotransferase isoform 2, and Chain A, Substrate Induced Remodeling Of The Active Site Regulates HTRA1 Activity.	Chun et al. (2016)	
MiR-21, MiR-155, and CD163	The neuroblastoma primary tissue samples.
The neuroblastoma (NBL) cell lines CHLA-255, LA-N-1, SK-N-BE(2), KNCR, and IMR-32 cells.
Female nu/nu mice (5 weeks of age).	qRT-PCR analysis, Flow cytometry assay, and Luciferase reporter assay	The result indicated a new exosomic miR-21/TLR8/NF-κB/exosomic miR-155/TERF1 axis triggered regardless of M1- or M2- polarization, but not in dendritic cells involved in resistance to chemotherapy in NBL, and identifies exosomes within the TME as important molecular targets to restore drug sensitivity.	Challagundla et al. (2015)	
Major histocompatibility complex II (MHC II), Hsp90 and flotillin-1	The human neuroblastoma cell line SH-SY5Y cells.
The human melanoma cell line A375 cells.	Electron microscopy assay, and Western blotting	The SH-SY5Y neuroblastoma-derived exosomes comprised of MHC II, Hsp90 and flotillin-1, whereas other cargo proteins or neuron specific proteins, such as actin or tau, NeuN, Sv2, are not released. Moreover, the results showed that, when applied extracellularly, exosomes released from neuronal cells modulated differentiation of melanoma cells.	Park, Ahn & Kim (2015)	

We thank Masoud Abak for the exact designing of figures that considerably improved the quality of this manuscript.

Additional Information and Declarations

Competing Interests

Author Contributions

Data Availability

The authors declare there are no competing interests.

Atefe Abak, Alireza Abhari and Sevda Rahimzadeh conceived and designed the experiments, performed the experiments, analyzed the data, contributed reagents/materials/analysis tools, prepared figures and/or tables, authored or reviewed drafts of the paper, approved the final draft.

The following information was supplied regarding data availability:

The research in this article did not generate any data or code.

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
