# Peer review of "Exosomes in cancer: small vesicular transporters for cancer progression and metastasis, biomarkers in cancer therapeutics"

_PeerJ, doi:10.7717/peerj.4763_

## Round 0.1 · original submission · Minor Revisions

Dear Atefe,

Two reviewers knowledgeable in the field of exosome research and cancer found your review to be comprehensive and a valuable contribution to PeerJ and the scientific community. Please address the minor revisions requested regarding figure legends, grammar and typos. I look forward to your resubmission.

Thomas Sanderson

·

Basic reporting

The review by Atefe Abek and colleagues is a comprehensive attempt to describe the multifaceted roles of microvesicles, exosomes, and microparticles in the carcinogenesis process. The authors cover a vast array of reports that have described the roles of these tiny vesicles in cancer progression, metastasis and drug resistance. The article is supported by a comprehensive list of updated references. The 5 figures are quite high quality and capture the essence of the exosomal signaling related to cancer. The table is very comprehensive and covers the subject in its entirety.

Specific comments:

The legends to the figure can be enhanced. This would make the figures self-explanatory

There are several instances where the authors have mixed up the tense which should be checked and corrected. For example the use of had is misplaced in this article

Typos should be checked and corrected

Experimental design

The review is very comprehensive and well designed

Validity of the findings

The article text is supported by updated references

Additional comments

The review by Atefe Abek and colleagues is a comprehensive attempt to describe the multifaceted roles of microvesicles, exosomes, and microparticles in the carcinogenesis process. The authors cover a vast array of reports that have described the roles of these tiny vesicles in cancer progression, metastasis and drug resistance. The article is supported by a comprehensive list of updated references. The 5 figures are quite high quality and capture the essence of the exosomal signaling related to cancer. The table is very comprehensive and covers the subject in its entirety.

Specific comments:

The legends to the figure can be enhanced. This would make the figures self-explanatory

There are several instances where the authors have mixed up the tense which should be checked and corrected. For example the use of had is misplaced in this article

Typos should be checked and corrected

Reviewer 2 ·

Basic reporting

I think this paper contains a serious concern regarding the paper which they have cited. For instance, some of the paper, which they have cited, have used ExoQuck, which is the commercially available kit for isolation of "exosome" with contamination of other soluble proteins.
Authors should exclude these kinds of paper, which are not able to provide the evidence that each paper really worked on the exosome or not.

Experimental design

No comment on this. This paper is Review paper and there is no new finding based on the experiment.

Validity of the findings

No comment on this. This paper is Review paper and there is no new finding based on the experiment.

Additional comments

Overall, this manuscript was well written regarding the roles of the exosome in cancer development, however, some of the content of some of the topic is a little superficial. For instance, "6. Exosomes Isolation and Analysis" are critical for exosome research, and there are huge issues to be solved for this. Authors should describe the pros and cons of each method and should mention to the reader that commercially available kit has some limitation for the analysis of exosome research. In addition, as I wrote in the above, the paper, which authors have cited, is not appropriate for me. For instance, when they have mentioned about pre-metastatic niche, they have cited Khalyfa et al rather than the following paper which showed the evidence regarding the contribution of exosome in pre-metastatic niche formation firstly in the world. Authors should check all of the references again and cite the paper which has shown the things firstly in that research field or follow the context of their manuscript.


Peinado H, Alečković M, Lavotshkin S, Matei I, Costa-Silva B, Moreno-Bueno G,
Hergueta-Redondo M, Williams C, García-Santos G, Ghajar C, Nitadori-Hoshino A,
Hoffman C, Badal K, Garcia BA, Callahan MK, Yuan J, Martins VR, Skog J, Kaplan
RN, Brady MS, Wolchok JD, Chapman PB, Kang Y, Bromberg J, Lyden D. Melanoma
exosomes educate bone marrow progenitor cells toward a pro-metastatic phenotype
through MET. Nat Med. 2012 Jun;18(6):883-91. doi: 10.1038/nm.2753.

---

## Round 0.2 · accepted · Accept

I have accepted the revised version as its contents has been adapted very well according to the reviewer comments and will contribute to PeerJ.. Thank you for your contribution.
Thomas